

# Subseasonal Midlatitude Prediction Skill Following QBO-MJO Activity

Kirsten J. Mayer [1] and Elizabeth A. Barnes [1]

[1]Department of Atmospheric Science, Colorado State University, Fort Collins, CO, USA.

**Correspondence:** Kirsten J. Mayer (kjmayer@rams.colostate.edu)

**Abstract.** The Madden-Julian Oscillation (MJO) is known to force extratropical weather days-to-weeks following an MJO event through excitation of stationary Rossby waves, tropical-extratropical teleconnections. Prior research has demonstrated that this tropically forced midlatitude response leads to increased prediction skill on subseasonal to seasonal (S2S) timescales. Furthermore, the Quasi-Biennial Oscillation (QBO) has been shown to possibly alter these teleconnections through modulation of the MJO itself and the atmospheric basic state upon which the Rossby waves propagate. This implies that the MJO-QBO relationship may affect midlatitude circulation prediction skill on S2S timescales. In this study, we quantify midlatitude circulation sensitivity and prediction skill following active MJOs and QBOs across the Northern Hemisphere on S2S timescales through an examination of the 500 hPa geopotential height field. First, a comparison of the spatial distribution of Northern Hemisphere sensitivity to the MJO during different QBO phases is performed for ERA-Interim reanalysis and ECMWF and NCEP hindcasts. Secondly, differences in prediction skill in ECMWF and NCEP hindcasts are quantified following MJO-QBO activity. We find that regions across the Pacific, North America and the Atlantic exhibit increased prediction skill following MJO-QBO activity, but these regions are not always collocated with the locations most sensitive to the MJO under a particular QBO state. Both hindcast systems demonstrate enhanced prediction skill 7-14 days following active MJO events during strong QBO periods compared to MJO events during neutral QBO periods.

## 1 Introduction

Previous research has focused on the impact of the Madden-Julian Oscillation (MJO) on the extratropical circulation in order to extend midlatitude prediction skill (e.g. Henderson et al., 2016; Baggett et al., 2017; Tseng et al., 2018; Zheng et al., 2018). The MJO is a 20-90 day tropical intraseasonal convective oscillation (Madden and Julian, 1971, 1972, 1994), and through its convective heating, initiates an extratropical response through the excitation of stationary Rossby waves. These waves modulate the mid-latitude circulation days to weeks following MJO activity and have been shown to provide coherent and consistent modulation of midlatitude circulation into subseasonal-to-seasonal (2-5 Weeks; S2S hereafter) timescales (e.g. Hoskins and Karoly 1981; Sardeshmukh and Hoskins 1988; Henderson et al. 2016; Tseng et al. 2018).

More recent research has demonstrated a dependence of the MJO on a stratospheric phenomenon known as the Quasi-biennial Oscillation (QBO). The QBO is an approximately 28 month, downward propagating zonal mean, zonal wind oscillation in the tropical stratosphere and has many subsequent impacts such as modulation of the upper tropical troposphere (e.g.



Collimore et al. 2003; Garfinkel and Hartmann 2011b; Son et al. 2017), the subtropical jet (e.g. Simpson et al. 2009; Garfinkel and Hartmann 2011a) and the stratospheric polar vortex (e.g. Holton and Tan 1980; Garfinkel et al. 2018). The QBO is typically divided into two phases, easterly and westerly (EQBO and WQBO, respectively), determined by the direction of the anomalous zonal wind in the lower tropical stratosphere (Baldwin and Dunkerton, 2001). Recent work has shown that the MJO convective
envelope tends to be stronger and have slower eastward propagation and longer path lengths during EQBO compared to WQBO (Son et al., 2017; Nishimoto and Yoden, 2017; Densmore et al., 2019; Zhang and Zhang, 2018). Son et al. (2017) hypothesize that this slower MJO propagation during EQBO is a consequence of strengthened MJO convection, as stronger MJO events tend to propagate more slowly across the Maritime Continent. However, Zhang and Zhang (2018) argue that stronger MJO wintertime events during EQBO are a consequence of a greater number of MJO days instead of larger amplitudes of individual
MJO events. While there are still uncertainties regarding the exact impacts of the QBO on the MJO, these studies demonstrate the importance of considering the QBO in MJO research.

Much of the recent MJO-QBO research has focused on the direct impacts of the QBO on the tropical tropopause, and thus, MJO activity, while only a handful of studies have examined how the QBO subsequently impacts MJO teleconnections (Baggett et al., 2017; Mundhenk et al., 2018; Wang et al., 2018). Baggett et al. (2017) and Mundhenk et al. (2018) emphasize
the impact of the QBO on MJO teleconnections through its modulation of MJO-induced Rossby waves, and consequently, changes in the steering and frequency of atmospheric rivers. Wang et al. (2018) found that when accounting for the phase of the QBO, the amplitude of the North Pacific storm track shift in response to MJO activity is greater during EQBO compared to WQBO, which they hypothesize to be from increased MJO strength during EQBO.

An MJO-QBO relationship has also been found in dynamical models. For example, Abhik and Hendon (2019) recently
demonstrated that hindcast simulations, initialized with observations during active MJOs, capture the increase in MJO amplitude and maintenance during EQBO events after about 5 days. In addition, this strengthened MJO amplitude during EQBO has been shown to translate to increased MJO prediction skill (Marshall et al., 2017; Lim et al., 2019), suggesting that the prediction skill of the subsequent midlatitude teleconnections may also increase following the MJO under EQBO conditions. Baggett et al. (2017) further show that S2S prediction skill of atmospheric rivers is increased within ECMWF hindcasts over
North America out to 3 weeks following MJO activity. This highlights the potential for an MJO-QBO relationship to modulate midlatitude prediction skill on S2S timescales.

Since hindcast models capture the increase in MJO amplitude during EQBO as well as exhibit enhanced prediction skill of the MJO in Weeks 1-3 under strong QBOs, this raises the question as to whether the MJO-QBO relationship also translates to enhanced prediction skill of MJO teleconnections under specific QBO phases. This paper explores this question through
an analysis of the influence of the QBO on midlatitude prediction skill following active MJOs on S2S timescales within the ECMWF and NCEP hindcasts.



## 2  Data and Methodology

### 2.1  Data

We utilize daily mean 500-hPa geopotential height (z500; years 1979-2017) from the European Centre for Medium-Range
Weather Forecasts Interim reanalysis (ERA-I; Dee et al. 2011) as well as the ECMWF and NCEP hindcasts obtained from the
S2S database (Vitart, 2017) established by the World Weather Research Program/World Climate Research Program (WWRP/WCRP).
The ECMWF hindcasts are composed of 11 ensemble members with hindcasts initialized 4 times a week (years 1995-2016).
The NCEP hindcasts are composed of 4 ensemble members with hindcasts initialized daily (years 1999-2010). The different
number of members between ECMWF and NCEP may contribute to differences in results between models. In the following
analysis, the ensemble mean for both models was used.

We focus on December, January and February (DJF) since MJO teleconnections are strongest during boreal winter (e.g.
Madden 1986), and the relationship between the MJO and QBO is strongest during these months as well (e.g. Yoo and Son
2016; Son et al. 2017). The annual cycle is removed from the ERA-I reanalysis by subtracting the daily climatology of z500
across 1979-2017 from the z500 field. For the hindcast models, a daily, lead-dependent climatology is subtracted from each
models' z500 field. To do this, we calculate the daily climatology for each lead time independently. Since the ECMWF model
is not initialized daily, two (forward and backward moving) 31-day running means are applied to the climatology at all lead
times to reduce noise, following Sun et al. (2018). These smoothed lead-dependent daily climatologies are then subtracted
from the z500 field of the corresponding model to remove the annual cycle.

There is presently no definitive understanding of the impact of the El Nino Southern Oscillation (ENSO) on the QBO-MJO
relationship. Some earlier research indicates that ENSO has a limited impact on the QBO-MJO interaction (e.g. Yoo and Son
2016; Nishimoto and Yoden 2017); however, recent work on QBO-MJO teleconnections has shown a possible dependency
of results on ENSO (Son et al., 2017; Wang et al., 2018; Sun et al., 2019). Thus, in an attempt to ensure our results are not
somehow biased by ENSO, we use the Nino3.4 Index (climatedataguide.ucar.edu/climate-data) to remove strong ENSO winter
seasons from our analysis. Specifically, when the amplitude of the NINO3.4 index for a month within DJF is greater than 1°C
(signifying El Nino) or less than -1°C (signifying La Nina), that DJF season is excluded from the analysis. With that said, we
have repeated our analysis with ENSO seasons included and find our conclusions remain the same (see Supplemental Figures
S7-S10).

### 2.2  MJO and QBO Indices

The real-time multivariate MJO (RMM) index is used to define the amplitude and phase of the MJO in the ERA-I reanalysis
(Wheeler and Hendon, 2004). This index uses empirical orthogonal function (EOF) analysis applied to anomalous outgoing
longwave radiation (OLR) and 200- and 850-hPa zonal wind, near-equatorially averaged (15°S to 15°N), to determine the first
two principal components (RMM1 and RMM2). A day is considered to have an active MJO when the RMM amplitude for that
day (defined as $\sqrt{(RMM1^2 + RMM2^2)}$) is greater than 1.0. The MJO phase is then defined as $tan^{-1}(RMM2/RMM1)$
and largely corresponds to the longitudinal location of the convective envelope. Active MJO dates within ERA-I that correspond





to initialization dates in ECMWF and NCEP are determined from this index. The RMM index is not separately calculated for each hindcast model because we do not aim to quantify the ability of the models to forecast the MJO directly (e.g. Vitart 2017). Rather, we use the index calculated from reanalysis to see how the hindcast models initialized on observed active MJO days ultimately forecast MJO teleconnections.

Identical to the definition of (Yoo and Son, 2016), the QBO index is calculated within ERA-I using monthly standardized
zonal wind at 50-hPa, area-averaged between 10°S to 10°N. Westerly QBO (WQBO) and Easterly QBO (EQBO) events are defined as when the standardized value is greater than $0.5\sigma$ or less than $-0.5\sigma$, respectively. Absolute values less than $0.5\sigma$ are considered neutral QBO (NQBO) events.

## 2.3   Methods

Quantification of each models' ability to represent MJO teleconnections under different QBO phases is conducted using the
Sensitivity to the Remote Influence of Periodic Events (STRIPES) index (Jenney et al., 2019). STRIPES is an index recently developed to determine regions of extratropical sensitivity to remote periodic events such as the MJO. As used here, the STRIPES index quantifies the strength and consistency of MJO teleconnections in z500 through average phase and 0-28 day lead information at individual grid points for a variety of observed phase speeds (5-8 days/phase; Wheeler and Hendon 2004). Specifically, a composite of average z500 anomalies for each MJO phase and lead is created for each grid point in the Northern
Hemisphere. If a region is sensitive to the MJO, we expect alternating z500 anomaly stripes sloped at the phase speed of the MJO in the phase versus lead diagram (as seen in Supplemental Figure S1 for example). Regions not sensitive to the MJO will appear noisy with smaller amplitudes and less coherent stripes. Averages along the slopes corresponding to the MJO phase speed are calculated, and if there are alternating stripes (i.e. sensitivity to the MJO), the resultant vector will look like a sine wave, for which the amplitude can be calculated. The amplitude of this oscillatory vector is the STRIPES index (Jenney et al.
2019). Therefore, the more sensitive the region is to MJO teleconnections, the larger the STRIPES index.

Since our application focuses on extratropical sensitivity in z500, we do not standardize our data for STRIPES as in Jenney et al. (2019). Standardization may mute the extratropical signal due to the greater variability of z500 in the midlatitudes, which is of main interest here. For equal comparison of STRIPES between the models and reanalysis, we calculate STRIPES for ERA-I only with dates that overlap with the hindcasts. Thus, the ERA-I STRIPES figures differ for ECMWF versus NCEP
dates.

STRIPES values that are statistically larger than expected by chance are determined using the bootstrapping method. The number of random days grabbed corresponds to the observed number of days for the QBO-MJO event of interest. In order to retain autocorrelation within MJO events, we keep the day-of-year (DOY) and phase distribution information for each MJO event and randomly sample years (with replacement). Since the ECMWF hindcast data is not initialized on the same day each
year, if the DOY needed is not available for a particular year, we instead use the date of initialization closest to this DOY. From this sample, we calculate STRIPES. This is repeated 250 times for each latitude and longitude. We repeat this calculation 250 times due to computational limits. Any STRIPES value greater than the 90[th] percentile of these bootstrapped values are deemed significant. Since autocorrelation is retained, this statistical analysis is more difficult to pass, and thus, the 90[th] percentile was





used instead of the $95^{th}$ percentile. When the data is subdivided by QBO phase, we begin to see the effects of sample size on
the uncertainty, leading to fewer points of significance. However, when all MJO days are included (see Figure 3), the statistical
analysis shows significance in regions of large STRIPES values. This bootstrapping analysis is only conducted on ERA-I, as
these are the 'observed' sensitivities and thus, the regions of interest.

To quantify midlatitude prediction skill, a daily area-weighted Pearson correlation is conducted between hindcast and ERA-I
anomalous z500 (anomaly correlation coefficient; ACC). The data is separated into NQBO-, EQBO- and WQBO-MJO events
in each hindcast dataset and the corresponding reanalysis data is obtained from ERA-I. The ACC between a given model day
and the same day in ERA-I is calculated within a centered 60° longitude wide box extending from 30-60° N. Our conclusions
are not affected by the latitudinal extent of the box when it is varied by +/- 10-30 °N. This calculation is repeated for every ini-
tialization and subsequent lead time as well as every 5° longitude beginning at 0°E. ACCs are grouped and averaged by QBO
phase to obtain average ACCs across the Northern Hemisphere at every lead for each QBO phase (see Supplemental Figure
S2 for an example). Differences between EQBO- or WQBO-MJO ACCs and NQBO-MJO ACCs capture the additional mid-
latitude prediction skill following active MJOs during E/WQBO compared to neutral QBO. Differences between EQBO-MJO
or WQBO-MJO ACCs and EQBO-inactive MJO or WQBO-inactive MJO ACCs capture the additional midlatitude prediction
skill following active MJOs during a particular strong phase of the QBO (see Supplemental Table S1 for sample sizes).

Statistically significant differences in ACCs across lead and longitude are also computed with the bootstrapping method.
Specifically, all model data within DJF is shuffled and random dates are grabbed. The number of random dates corresponds to
the number of observed dates for the particular QBO phase and MJO activity being tested. The corresponding random dates
are then found in ERA-I. The spatial correlations between the model and the observations are calculated and then averaged to
get an average ACC. This is repeated for each QBO-MJO combination, and the differences between their ACCs is calculated.
The above analysis is repeated 10,000 times for each longitude and lead time. Differences greater than the $97.5^{th}$ percentile
of the 10,000 bootstrapped differences are considered significantly greater from that expected by chance. In this bootstrapping
analysis, we were able to repeat the calculations 10,000 times (instead of 250) because the calculation was less computationally
expensive.

## 3 Results

### 3.1 Extratropical Sensitivity

The left column of Figure 1 shows the STRIPES analysis of ERA-I for days within the ECMWF hindcasts, split by QBO
phase. Darker shading indicates regions of greater sensitivity to the MJO for each QBO state. Regions along the North Pacific
and Atlantic storm tracks as well as over North America are highlighted by STRIPES following the MJO for all phases of
the QBO (Figure 1a,c,e). This is consistent with previous research as these regions have been shown to be sensitive to MJO
excited Rossby waves through, for example, their modulation of the North Atlantic Oscillation (Cassou, 2008), the Pacific
North American Oscillation (Mori and Watanabe, 2008) and Northern Hemisphere wintertime blocking (Henderson et al.,
2016).





The right column of Figure 1 shows the STRIPES analysis of the ECMWF hindcasts for the same dates. ECMWF largely captures the spatial patterns and locations sensitive to the MJO under different QBO phases (spatial correlation with ERA-I: $r_{NQBO-MJO} = 0.92$, $r_{EQBO-MJO} = 0.93$, and $r_{WQBO-MJO} = 0.95$), but overall the model has smaller STRIPES values than ERA-I. This is likely a result of model forecast degradation at later lead times since the calculation of STRIPES utilizes z500 forecasts out to 28 days lead time.

An examination of the NCEP hindcasts shows that it also generally captures regions sensitive to the MJO under varying phases of the QBO (Figure 2b,d,f; spatial correlation with ERA-I: $r_{NQBO-MJO} = 0.96$, $r_{EQBO-MJO} = 0.95$, and $r_{WQBO-MJO} = 0.93$) and is also weaker than the corresponding ERA-I analysis (Figure 2a,c,e). The ERA-I STRIPES analysis for NCEP hindcasts largely has the same features as the ERA-I analysis for ECMWF hindcasts, but with larger values due to differences in sample size and dates of initialization between NCEP and ECMWF. From this STRIPES comparison (Figures 1 and 2), we conclude that the ECMWF and NCEP hindcast models generally capture Northern Hemisphere regions sensitive to the MJO as highlighted by large spatial correlations between each model and ERA-I.

Recent research has shown that during EQBO, the MJO amplitude is larger and the convective envelope propagates slower compared to MJO activity during WQBO (Son et al., 2017; Nishimoto and Yoden, 2017; Zhang and Zhang, 2018). If direct impacts to the MJO (e.g. through changes in upper tropospheric tropical static stability) lead to changes in MJO teleconnection sensitivity across the Northern Hemisphere, we might expect EQBO-MJO events to have larger midlatitude sensitivity to the MJO compared to WQBO-MJO. Based on our STRIPES analysis, we find that Northern Hemisphere sensitivity to the MJO is significantly reduced during EQBO-MJO events compared to WQBO-MJO events (compare Figure 1c,e and Figure 2c,e; significance not shown). We explored this further and found that this difference can be explained by the tendency for WQBO to have larger magnitude z500 anomalies compared to EQBO, not more distinct stripes, which is likely due to differences in sample size. In other words, when the amplitude differences between the z500 anomalies are accounted for through normalization, the difference in Northern Hemispheric sensitivity to the MJO between QBO phases is greatly reduced (Figure 3). The data is normalized by dividing by the average absolute value of the Phase vs Lead diagram for each latitude-longitude point prior to computing the STRIPES index. By doing so, we are able to reduce the impact of the anomaly magnitude on the STRIPES index, and thus, the index mainly provides information on the "stripey-ness".



**Figure 1.** STRIPES values for (left) ERA-Interim and (right) ECMWF for all (top) NQBO-MJO, (middle) EQBO-MJO and (bottom) WQBO-MJO events. (a,c,e) Black hatches denote STRIPES values that are statistically larger than expected by chance at 90% confidence in ERA-I.







**Figure 2.** STRIPES values for (left) ERA-Interim and (right) NCEP for all (top) NQBO-MJO, (middle) EQBO-MJO and (bottom) WQBO-MJO events. (a,c,e) Black hatches denote STRIPES values that are statistically larger than expected by chance at 90% confidence in ERA-I.

(a) ERA-I: EQBO-MJO

(b) ERA-I: EQBO-MJO

(c) ERA-I: WQBO-MJO

(d) ERA-I: WQBO-MJO

**Figure 3.** Normalized STRIPES values for (left) ECMWF hindcasts' dates in ERA-I and (right) NCEP hindcasts' dates in ERA-I for (top) EQBO-MJO and (bottom) WQBO-MJO events. Data is normalized by dividing by the average absolute value of the Phase vs Lead diagram for each latitude-longitude point and then calculating STRIPES on these normalized values.

## 3.2 Prediction Skill

### 3.2.1 Regional Prediction Skill

Knowing that the ECMWF and NCEP hindcasts generally capture regional sensitivity to the MJO, we next address whether the QBO impacts midlatitude skill during MJO events and whether regions of increased sensitivity to MJO-QBO activity translate to increased prediction skill. Here, skill is calculated as an anomaly spatial correlation between z500 from the hindcasts and ERA-I (see Section 2.3), and we compare this skill over active QBO-MJO combinations to skill during NQBO-MJO and inactive MJO. As mentioned in the introduction, EQBO has been found to impact the MJO in ways that may enhance MJO





teleconnections (e.g. Son et al. 2017; Nishimoto and Yoden 2017). Since enhanced activity may provide a prominent signal

above model noise and uncertainty, and thus, hypothetically lead to enhanced prediction skill, we focus here on only improved prediction skill (see Supplemental Figures S4-S5 for regions of decreased prediction skill).

Figure 4 shows z500 anomaly prediction skill as a function of lead time for the North Pacific (165°W, 30-60°N), North Atlantic (30°W, 30-60°N), and Europe regions (0°E, 30-60°N). There are multiple ways to think about skill following MJO-QBO activity and therefore we include two types of statistical information. The first type of significance (hollow circles)

represents the impact of the phase of strong QBOs on prediction skill compared to neutral QBO during active MJO. In other words, where the orange/teal line (EQBO-/WQBO-MJO) is significantly above the black line (NQBO-MJO). The second type of significance (colored dots) represents changes in prediction skill following active MJOs compared to inactive MJOs during a particular QBO phase, or said another way, where solid lines (EQBO-/WQBO-/NQBO-MJO) are significantly above the dashed lines (EQBO-/WQBO-/NQBO-noMJO). The presence of both of these forms of significance (colored dots within the

hollow circles) represents where there is greater prediction skill following active MJOs compared to inactive MJOs during a particular QBO phase (colored dots) *and* active MJOs during a strong QBO phases compared to active MJOs during NQBO (hollow circles). When these two significances appear together, we can say that a particular strong QBO increases the impact of the MJO on midlatitude prediction skill.

First we focus on the differences in skill between strong QBO phases and NQBO following active MJOs (hollow circles).

For ECMWF, the North Atlantic and Europe (Figure 4c,e) have significantly increased prediction skill out to Week 4 following WQBO-MJO (hollow circles on solid teal line) compared to NQBO-MJO. For NCEP, there is significantly increased prediction skill into Week 2 following EQBO-MJO in the North Pacific and scattered significance within Weeks 2, 3 and 4 in the North Atlantic and Week 4 over Europe following WQBO-MJO (Figure 4b,d,f).

Focusing next on differences in skill between active and inactive MJOs during strong QBO phases (colored dots), the MJO

leads to enhanced prediction skill compared to inactive MJO in the North Atlantic and Europe during WQBO out to Week 4 in ECMWF (Figure 4c,e; teal dots), and in Weeks 3-4 in the North Atlantic and Europe during WQBO in NCEP. For all of these cases, this increase in prediction skill following the MJO is not present during NQBO (absence of black dots), and suggests that the changes to the basic state and/or to the MJO itself during WQBO is associated with enhanced midlatitude MJO impact over the North Atlantic and Europe for ECMWF and NCEP.

The presence of both of these forms of significance (colored dots within the hollow circles) represents where a particular strong QBO increases the impact of the MJO on midlatitude prediction skill. In the three regions depicted in Figure 4, the two forms of significance overlap in ECMWF and NCEP over the North Atlantic and Europe through Week 3 and 4 (Figure 4c,e; teal dots inside hollow circles).





**Figure 4.** Anomalous spatial correlation coefficient at (top) 165°W, (middle) 30°W and (bottom) 0°E for (left) ECMWF and (right) NCEP. Solid lines correspond to active MJOs while dashed lines correspond to inactive MJOs. Colors refer to the phase of the QBO. Colored dots denote significantly increased skill between active and inactive MJO under a specific QBO state at 95% confidence. Hollow black circles indicate a significantly increased skill between E/WQBO-MJO events and NQBO-MJO events at 95% confidence.



### 3.2.2 Northern Hemisphere Prediction Skill: Dependence on active MJO

While Figure 4 shows results for three specific regions, we extend these results to all longitudes in Figures 5 and 6. To determine how the impact of the MJO on midlatitude prediction skill changes following a particular phase of the QBO (colored dots in Figure 4), we examine the difference between prediction skill following active MJOs compared to inactive MJOs during both EQBO and WQBO (Figure 5). Specifically, the four panels show the difference in ACC between EQBO-MJO and EQBO-noMJO (Figure 5a,b; orange solid and dashed lines in Figure 4) and WQBO-MJO and WQBO-noMJO (Figure 5c,d; teal solid 225 and dashed lines in Figure 4). The left column of Figure 5 shows the differences within ECMWF and the right column shows differences within NCEP. Shading specifies increased prediction skill following the MJO compared to inactive MJOs during the specific phase of the QBO. Regions of significantly increased prediction skill following the MJO compared to inactive MJOs during the specific phase of the QBO are denoted with grey dots (orange and teal dots in Figure 4).

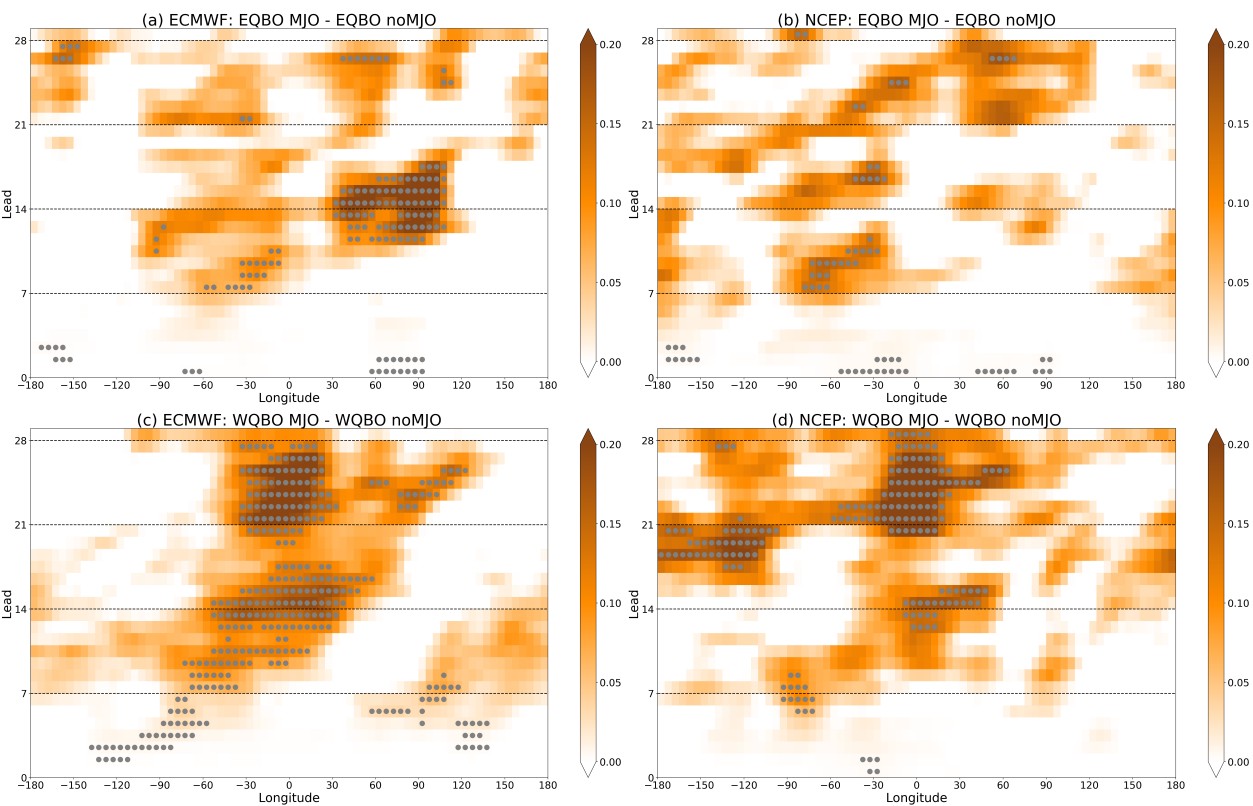

**Figure 5.** Anomalous correlation coefficient between (top) EQBO-MJO and EQBO-noMJO and (bottom) WQBO-MJO and WQBO-noMJO for (left) ECMWF and (right) NCEP at each longitude and lead from model initialization. Correlations are calculated within a 60° wide box, centered on each longitude, extending from 30-60°N. Gray dots denote significant increases in prediction skill at 95% confidence from active MJO compared to inactive MJO under the specific QBO phase for the plot.



During EQBO in both models (Figure 5a,b), there is enhanced prediction skill following active MJOs starting in North
America and extending into Asia (90°W - 90°E) at Week 2-3 leads for ECMWF and at Week 2 leads for NCEP over North
America. During WQBO in ECMWF (Figure 5c), there is increased prediction skill following the MJO in the East Pacific
into North America through Week 1. In ECMWF, this increased skill continues through Week 2 into the North Atlantic and
continues over the Atlantic and Europe from Week 3-4 (Figure 5c). In NCEP, additional prediction skill occurs in the Pacific
during Week 3 and over the North Atlantic by Week 4 (Figure 5d).

From Figure 5 we see that in both models, active MJOs during EQBO generally lead to enhanced skill from North America
to East Asia while active MJOs during WQBO generally lead to enhanced skill from the North Pacific through Europe on
subseasonal timescales. The regions of enhanced prediction skill following active MJOs during EQBO and WQBO are not
associated with enhanced prediction skill following active MJOs during NQBO (see Supplemental Figure S6). This suggests
that following MJO activity, subseasonal prediction skill is enhanced in the Pacific to Europe by the MJO during strong QBOs
while MJO activity during NQBO does not significantly enhance prediction skill (although sample size for NQBO-noMJO is
small, Table S1).

### 3.2.3   Northern Hemisphere Prediction Skill: Dependence on active QBO

To further explore the importance of QBO-MJO activity on subseasonal prediction, we examine the difference between pre-
diction skill following active MJOs during strong QBO compared to active MJOs during NQBO. Similar to Figure 5, the left
column of Figure 6 shows the differences for ECMWF and the right column for NCEP. Specifically, the four panels show the
difference in ACC between EQBO-MJO and NQBO-MJO (Figure 6a,b; orange and black solid lines in Figure 4) and WQBO-
MJO and NQBO-MJO (Figure 6c,d; teal and black solid lines in Figure 4). As in Figure 4, black hollow circles indicate
significant increases in prediction skill between the specified QBO and NQBO during active MJO.

During EQBO in both models (Figure 6a,b), there is mainly enhanced prediction skill following active MJOs over the
Pacific in Week 1 and over North America in Week 2 compared to NQBO. For WQBO in both models (Figure 6c,d), there is
also enhanced prediction skill following active MJOs compared to NQBO from Week 1 to 4 over the Pacific and into Europe.
Specifically, this enhanced prediction skill in ECMWF (Figure 6c) is located in the Pacific and extends into the Atlantic for
Weeks 1-2, and continues through the Atlantic and into Europe during Weeks 3 and 4. In NCEP (Figure 6d), this enhanced
prediction skill spans from the Pacific to Europe during Weeks 1-4.

Note that in all panels, much of the enhanced skill is confined to a specific longitudinal region. Since the QBO oscillates
with a period of about 28 months, we may expect enhanced prediction skill to remain around the same region through Week 4
when examining skill differences between QBO phases. However, this confined skill could also be due to a stationary rossby
wave signal following strong QBO-MJOs that is not present following NQBO-MJOs. Therefore, since the prediction skill is
enhanced and confined to a specific longitudinal region out to Week 3, we speculate that this enhanced non-propagating skill is
likely from either a stationary rossby wave signal or enhanced skill from the strong QBOs effect on the midlatitudes compared
to NQBO.




Since EQBO is thought to increase the amplitude of the MJO as well as help to propagate the MJO further into the Pacific Ocean compared to WQBO (Son et al., 2017; Nishimoto and Yoden, 2017; Zhang and Zhang, 2018), it may be expected that active MJOs during EQBO conditions will lead to stronger MJO teleconnections and thus, act to enhance subseasonal
prediction in the midlatitudes. However, from Figure 6 we see that both EQBO *and* WQBO tend to have greater prediction skill compared to NQBO during active MJO across a range of longitudes and lead times. Specifically over the Pacific during EQBO-MJO and the Pacific through Europe during WQBO-MJO. While unexpected, this result is partially supported by previous research, where enhanced prediction skill of Atmospheric Rivers over Alaska is found following active MJOs during WQBO (Baggett et al., 2017), and the North Atlantic Oscillation and MJO connection is stronger during WQBO (Feng and
Lin, 2019).

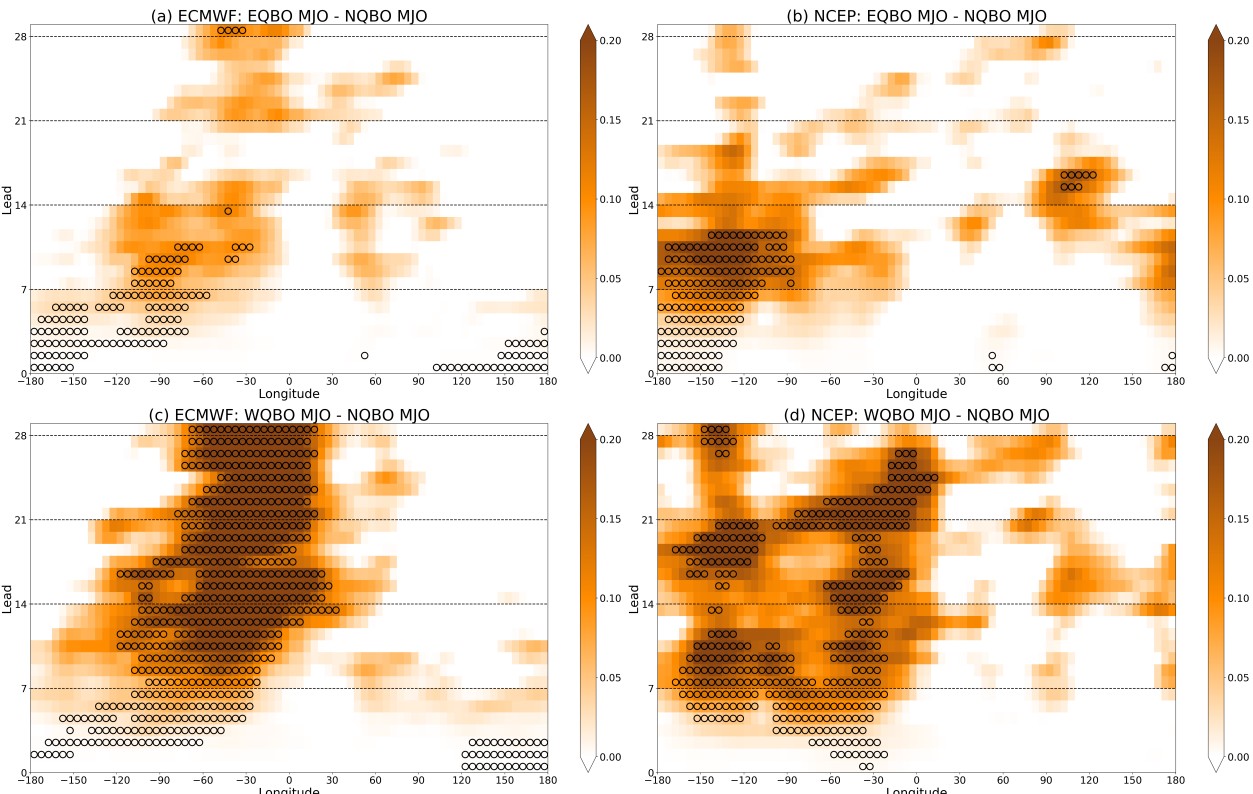

**Figure 6.** Anomalous correlation coefficient between (top) EQBO-MJO and NQBO-MJO and (bottom) WQBO-MJO and NQBO-MJO for (left) ECMWF and (right) NCEP at each longitude and lead from model initialization. Correlations are calculated within a 60° wide box, centered on each longitude, extending from 30-60°N. Hollow black circles indicate significant increases in prediction skill at 95% confidence from E/WQBO-MJO activity compared to NQBO-MJO activity.





### 3.2.4 Summary of Northern Hemisphere Prediction skill

The presence of both of these forms of significance, grey dots in Figure 5 and hollow circles in Figure 6, represents where strong QBOs increase the impact of the MJO on midlatitude prediction skill. As a reminder, hollow circles indicate where there is significantly greater skill following EQBO or WQBO-MJO than NQBO-MJO, while grey dots indicates where EQBO-

, WQBO-, or NQBO-MJO leads to significantly greater skill than EQBO-, WQBO-, or NQBO-noMJO. To better visualize this overlap, Figure 7 combines both forms of significance from Figures 5 and 6 for ease of visualization, where the previously grey dots are now orange (teal) for EQBO (WQBO). In EQBO in both models (Figure 7a,b), there is very little overlap of the two forms of significance (orange dots in hollow circles). On the other hand, for WQBO in both models (Figure 7c,d), most of the East Pacific and Atlantic that exhibit significantly increased prediction skill following active MJOs compared to inactive MJOs

(teal dots) are collocated enhanced prediction skill following WQBO-MJO compared to NQBO-MJO (hollow circles). This indicates that there is significantly greater prediction skill following active MJOs compared to inactive MJOs during WQBO as well as active MJOs during NQBO. In other words, WQBOs increase the impact of the MJO on midlatitude prediction skill between the East Pacific and Atlantic compared to NQBO.

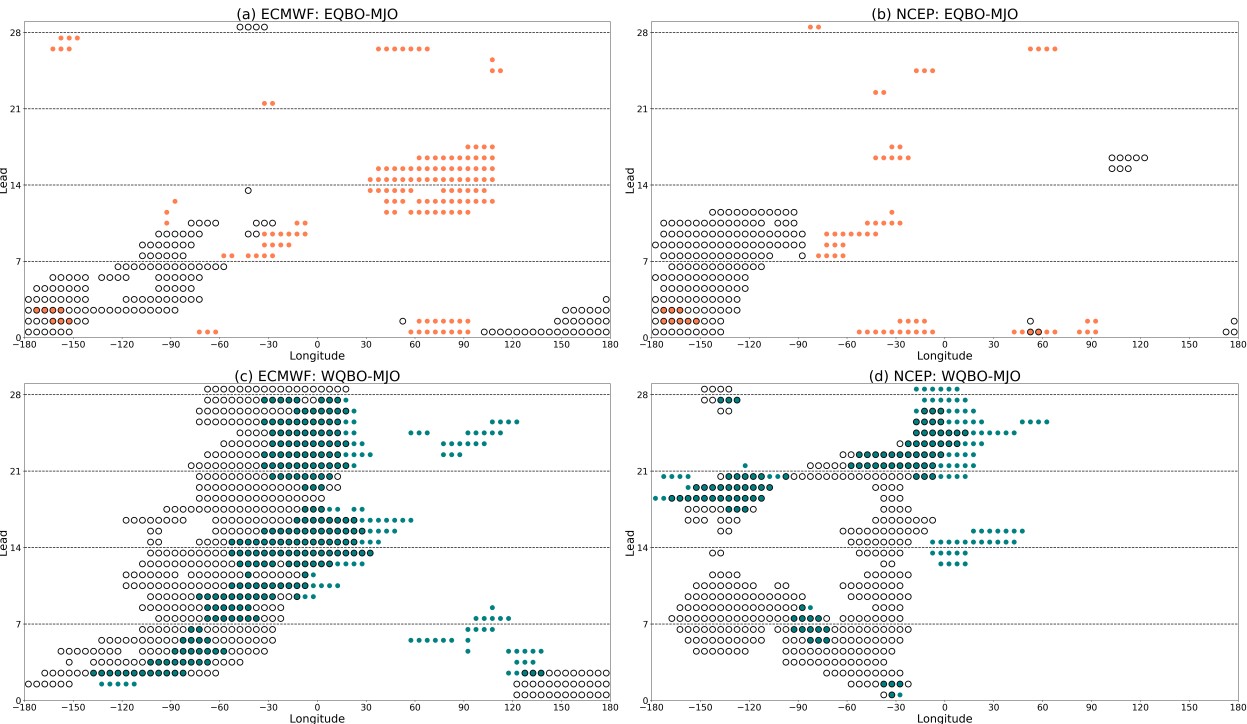

**Figure 7.** Lead vs longitude plots with combined significance from Figure 5 and 6. Colored dots denote significant increases in prediction skill at 95% confidence from active MJO compared to inactive MJO under the specific QBO phase for the plot, where the color refers to the phase of the QBO. Orange is EQBO and teal is WQBO. Hollow black circles indicate significant increases in prediction skill at 95% confidence from E/WQBO-MJO activity compared to NQBO-MJO activity.




### 3.2.5 Northern Hemisphere Prediction Skill and Sensitivity

In section 3.1, we saw that ECMWF and NCEP hindcasts generally capture regional sensitivity to the MJO under different phases of the QBO. From previous research, we also know that robust midlatitude circulation following certain phases of the MJO tends to have additional forecast skill (Tseng et al., 2018), and therefore, we may expect a link between regional sensitivities and increased prediction skill.

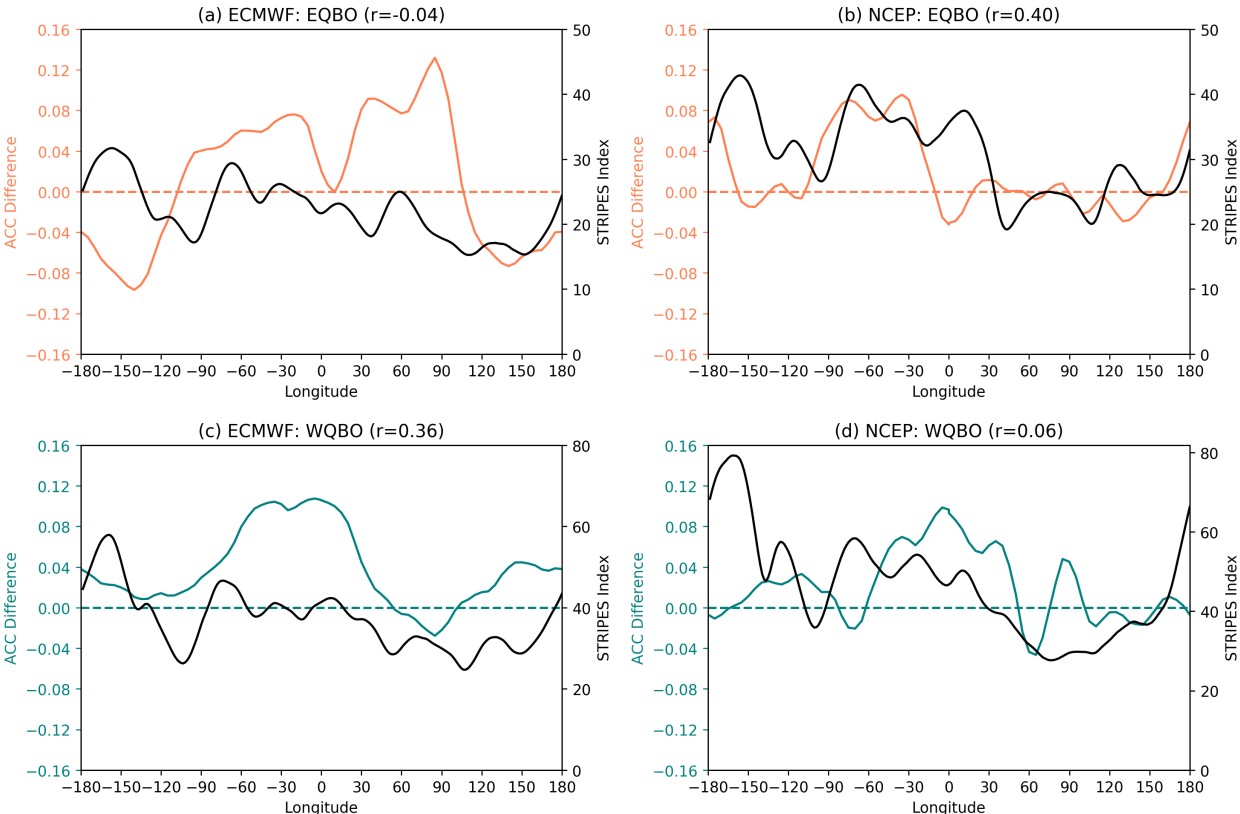

**Figure 8.** Average change in prediction skill between active and inactive MJOs (color) across leads of 8-18 days and average STRIPES value (black) from 30° to 60°N for all longitudes. Colors refer to the phase of the QBO, where orange is EQBO and teal is WQBO. The correlation coefficient (r) is given in the title.

In an attempt to systematically examine the relationship between MJO sensitivity and prediction skill across all longitudes,
STRIPES values are averaged from 30-60°N and compared to prediction skill averaged along leads 8-18 days (Figure 8). Days 8-18 are chosen based on previous research on MJO teleconnection timescales (e.g. Cassou 2008; Henderson et al. 2016; Tseng et al. 2019), however, these results are insensitive to variations of +/- 5 days. Figure 8 shows the average prediction skill across leads 8-18 days for EQBO in orange (Figure 8a,b) and WQBO in teal (Figure 8c,d) along with average STRIPES values in black for all longitudes. While one can certainly find locations where they appear to oscillate together, their correlations are





low (see panel titles). The exception being NCEP during EQBO (Figure 8b) and ECMWF during WQBO (Figure 8c), where the correlation is around 0.4. For the other two panels, it appears that increased regional z500 sensitivity to the MJO in the Northern Hemisphere does not clearly translate to increased prediction skill. It is possible that these correlations are low due to differences in the signal-to-noise ratio between composites and daily spatial correlations.

## 4    Conclusions

The MJO is the dominant mode of intraseasonal variability in the tropics (Madden and Julian, 1971; Adames and Kim, 2016), and through its convective heating, modulates midlatitude weather, days to weeks after an MJO event (e.g. Vecchi 2004; Zhou et al. 2012; Henderson et al. 2016; Tseng et al. 2019). Recent research has shown that the QBO impacts MJO amplitude, propagation, and prediction skill (Son et al., 2017; Nishimoto and Yoden, 2017; Zhang and Zhang, 2018; Marshall et al., 2017; Lim et al., 2019) as well as modulates MJO teleconnections (e.g. Baggett et al. 2017; Mundhenk et al. 2018; Wang et al. 2018).

This raises the question as to whether the QBO also affects the prediction skill of MJO teleconnections. The goal of this study is to address this question through an examination of differences in Week 1-4 prediction skill between different combinations of QBO-MJO activity.

Through a STRIPES analysis, we show that ECMWF and NCEP hindcasts are capable of simulating midlatitude MJO sensitivity, in a composite sense, out to Week 4 under different phases of the QBO. Thus, we use these hindcasts to study

enhanced S2S prediction skill following QBO-MJO activity. Increased prediction skill is determined from significant increases in spatial correlations of z500 for various QBO-MJO combinations. First, comparing strong QBOs to NQBOs, we find that there is enhanced prediction skill following MJOs during EQBO over the Pacific, and enhanced prediction skill from the Pacific to Europe following MJOs during WQBO. Second, comparing active MJOs to inactive MJOs during different QBO phases, we find that when active MJOs occur during EQBOs, there is enhanced prediction skill from North America into Asia over

Weeks 2-3 in ECMWF and Weeks 2-4 in NCEP. During WQBO, this enhanced prediction skill is located in the Pacific through North America in Week 1 and continues through Week 2 over the North Atlantic and through Week 3-4 over the Atlantic and Europe in ECMWF. Additional prediction skill in NCEP appears in the Pacific during Week 3 and the North Atlantic by Week 4. In contrast, there is no enhanced prediction skill following MJO activity compared to inactive MJOs during NQBO in these regions and suggests that the impact of the MJO on prediction skill over the Pacific to the Atlantic is only apparent during

strong QBOs.

Together, these two forms of significance inform us on when and where strong QBOs increase the impact of the MJO on midlatitude prediction skill. Over North America, the Atlantic and Europe (ECMWF and NCEP) following active MJOs during WQBO, the two forms of significance overlap and thus, implies that WQBO (compared to NQBO) increases the impact of the MJO on midlatitude prediction skill. On the other hand, regions with both forms of significance during EQBO are scarce. When

comparing all regions of enhanced prediction skill to regional sensitivity (STRIPES), we found no clear relationship, except possibly in ECMWF during WQBO and NCEP during EQBO.



This study provides insight on improved prediction skill following different MJO-QBO combinations; however, more research is needed to determine the causal link between the MJO-QBO, midlatitude teleconnections and prediction skill. It is unclear whether enhanced midlatitude prediction skill is a consequence of the QBO's direct effects on the tropical environment
in which the MJO forms and/or through the modulation of the atmospheric basic state through which Rossby waves propagate.

We motivate this study by suggesting that enhanced MJO prediction following EQBO (Marshall et al., 2017; Lim et al., 2019; Abhik and Hendon, 2019) may also lead to enhanced midlatitude prediction skill following MJOs during EQBO. However, we find that *both* EQBO and WQBO lead to enhanced midlatitude prediction skill in these hindcasts rather than only EQBO. Enhanced skill following MJOs during both EQBO and WQBO may partially be explained by Kim et al. (2019) who find no
significant impact of the QBO on MJO prediction skill within the SubX database. In addition, there is a growing body of work suggesting the importance of WQBO on MJO teleconnections. For example, recent work has shown that the North Atlantic Oscillation and MJO relationship is stronger during WQBO (Feng and Lin, 2019) and prediction skill of Atmospheric Rivers over Alaska is enhanced following WQBO-MJO (Baggett et al., 2017). Finally, while strong ENSO events were removed from our analysis in an attempt to separate the effects of the QBO from those of ENSO, an ENSO influence may still remain.
In addition, the sample sizes for MJO-QBO activity are not large (Table S1), although we attempt to account for this through statistical analysis. Even so, this work suggests that both phases of the QBO may impact prediction skill of MJO teleconnections and should be considered in future studies.

*Data availability.* ERA-I Reanalysis data are provided by the European Centre for Medium Range Forecasts (https://www.ecmwf.int/en/forecasts/-datasets/reanalysis-datasets/era-interim/). The ECMWF and NCEP hindcasts are provided by the World Weather Research Program/World
Climate Research Program (ECMWF: https://apps.ecmwf.int/datasets/data/s2s/ and NCEP: http://iridl.ldeo.columbia.edu/SOURCES/.Models/.SubX/) RMM Index data are provided by the Australian Bureau of Meteorology (http://www.bom.gov.au/climate/mjo/graphics/rmm.74toRealtime.txt). The Nino3.4 Index data was provided by NOAA/OAR/ESRL Boulder, Colorado, USA (climatedataguide.ucar.edu/climate-data/).

*Author contributions.* Kirsten Mayer led the collection of the data, the data analysis and writing of the manuscript. Elizabeth Barnes aided with the experimental design, analysis techniques and writing of the manuscript.

*Competing interests.* The authors declare that they have no competing interests.

*Acknowledgements.* This research was partially funded by NOAA Research through supporting K. J. M. and partially funded by the NOAA MAPP S2S Prediction Task Force through supporting E. A. B. on NOAA grant NA16OAR4310064.





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
