# Peer review of "Subseasonal Midlatitude Prediction Skill Following QBO-MJO Activity"

_Weather and Climate Dynamics, 2019_

## Referee Comment (RC1) · Anonymous Referee #1 · 20 Jan 2020

Review of " Subseasonal Midlatitude Prediction Skill Following QBO-MJO Activity" Author(s): Kirsten J. Mayer and Elizabeth A. Barnes

Recommendation: Major revisions

The authors analyze the possibility that the state of the QBO can lead to enhanced predictability of MJO teleconnections. After first demonstrating that the models capture in a gross sense the MJO teleconnections in the Pacific and Atlantic sector, and that the stripey-ness of the teleconnection is generally similar in all QBO phases, the authors compare the ACC for different combinations of active/inactive MJO and EQBO in their various phases. They find significantly higher ACC when both are active, with the increase in ACC even larger for WQBO as compared to EQBO.

[Figure]

I had trouble accepting the conclusions the authors reach from figures 4,5, and 6 regarding any interaction between the QBO and the MJO in their extratropical teleconnections. Rather, these figures seem to separately reflect skill added alternately by either the QBO or the MJO. Please see my general comment below. My other two major comments concern the accessibility of this paper to someone not already very familiar with the author's previous work, and they should be easier to address.

Major comments:

1. While I appreciate the power of the STRIPES analysis, I must point out that the first time I read the paper I did not understand at all what the authors were doing. Only after skimming Jenney et al 2019 and looking at supplemental figure 1 did I fully understand what was happening. I worry that a casual reader may be less patient. To be constructive, I suggest that supplemental figure 1 be included in the main text, and I would also suggest adding a figure of lat vs. lon Z500 with a few panels corresponding to different periods explicitly showing how the wave train leads to Z500 alternating anomalies. I realize this is already in Jenney et al but a new, at first not intuitive, index needs a certain amount of repetition. As as aside, I was surprised that the STRIPES was just as strong in the European sector as in North Pacific/ NorthAmerica. I would have expected a stronger response closer to the Pacific.

The ACC results also indicate that the additional predictability from the MJO is mainly in the Atlantic sector too rather than the North Pacific (Figures 4 and 5). To me this is counter-intuitive, as the MJO should immediately and directly affect the North Pacific, especially in the first few weeks, and then affect the Atlantic more weakly later on. Additional discussion would be helpful. (I can try to reason why my intuition is incorrect, but really the authors should help with this)

2. Between lines 192 and 203 the authors form an argument that I don't find convincing. As this argument underlies the reset of the paper, this is a major issue.

To this reviewer, the clearest evidence that the QBO can enhance MJO related pre-

diction skill would be if the difference in ACC between EQBO/MJO and EQBO/noMJO or between WQBO/MJO and WQBO/noMJO is larger than the difference between noQBO/MJO and noQBO/noMJO. Based on supplemental table 1 it seems that this kind of comparison isn't possible due to possible contamination by the ENSO signal, though perhaps the authors could compute the mean Nino3.4 index for each composite included on supplemental table 1. If the mean Nino3.4 value for each composite is small, then La Nina and El Nino events balance out and the net prediction skill added by ENSO is small.

Instead the authors evaluate a pair of differences that only partially reflect on whether the QBO is enhancing MJO related prediction skill, but rather reflect alternately on whether there is prediction skill associated with the MJO, and separately whether is prediction skill associated with the QBO (in Figures 4-6). Unless the authors perform the test in the previous paragraph, there is no basis for this statement of the authors "When these two significances appear together, we can say that a particular strong QBO increases the impact of the MJO on midlatitude prediction skill".

Stated another way, the difference EQBO/MJO minus noQBO/MJO does not reflect anything about the MJO per se. Rather it reflects skill associated with EQBO. Hence I don't find figure 6 useful, other than the fact that it shows that the QBO enhances skillful forecasts in the Atlantic sector (which is a nice result, and consistent with Garfinkel et al 2018 already cited and Boer and Hamilton 2008, but the authors interpretation is completely different). In order for Figure 6 to have any bearing on the MJO, the authors need to include an additional figure showing EQBO/noMJO minus noQBO/noMJO to which we can compare the difference shown in figure 6. If there is a significant difference between EQBO/MJO minus noQBO/MJO as compared to EQBO/noMJO minus noQBO/noMJO, then there is evidence that there is some mutual interaction between the MJO and the EQBO. The authors could then rinse and repeat for WQBO.

In its present form, the authors analysis only convinces me that both the QBO or the MJO separately enhance predictability on S2S timescales in these models as compared to noQBO or noMJO.

3. I found section 3.2.5 extraneous and hard to understand without first skimming Tseng et al 2018. Consider deleting.

Minor comments: Line 13 "7-14 days", actually there is enhanced predicatability up to day 28 in figures 4-6. Why limit to 14 days?

Line 77 There is earlier work that argues that the QBO may modulate ENSO teleconnections. See Garfinkel and Hartmann 2010, Richter et al 2015, and Hansen et al 2016 Technical comments Line 2 stationary Rossby wave **and** tropica-extratropical teleconnections

Line 19 excitation of **quasi**stationary Rossby waves (the MJO can't force stationary waves on monthly mean or seasonal mean timescales)

Line 126 the reference to figure 3 seems incorrect. Figure 3 shows something else entirely.

Figure 1, title of bottom-right panel is incorrect (It probably should be WQBO-MJO)

Garfinkel, C.I. and Hartmann, D.L., 2010. Influence of the quasi‐biennial oscillation on the North Pacific and El Niño teleconnections. Journal of Geophysical Research: Atmospheres, 115(D20).

Richter, J.H., Deser, C. and Sun, L., 2015. Effects of stratospheric variability on El Niño teleconnections. Environmental Research Letters, 10(12), p.124021.

Hansen, F., Matthes, K. and Wahl, S., 2016. Tropospheric QBO–ENSO interactions and differences between the Atlantic and Pacific. Journal of Climate, 29(4), pp.1353-1368.

Boer, G.J. and Hamilton, K., 2008. QBO influence on extratropical predictive skill. Climate dynamics, 31(7-8), pp.987-1000.

---

## Referee Comment (RC2) · Anonymous Referee #2 · 28 Jan 2020

This manuscript aims to assess the impacts of the QBO and MJO states on subseasonal prediction skills of midlatitude circulation. Interesting results are presented such as the difference in the sensitivity of prediction skill to the MJO state depending on the QBO state. I think this manuscript would further improve if the authors could provide more clarification to their methodology. I also suggest a few points for the authors to reconsider the interpretations and conclusions of the results below.

Comments

1. Section 2.3: I suggest the authors revise the method section to make it more accessible to a broader audience. The authors also jump into explaining the details of each analysis technique (i.e., STRIPES and ACC). Before jumping into the details, it would be helpful to the readers if the authors could first outline what they attempt to

quantify and how it relates to the objective of this study. More specifically, I suggest the following points.

a. For readers who are unfamiliar with Jenney et al. 2019, it would be difficult to understand the STRIPES index. I suggest to move the Supplemental Figure S1 to the main manuscript and include further visual illustrations on how the STRIPES index is calculated.

b. I suggest the authors add more discussion on the novelty and benefits of STRIPES analysis. Why do the authors choose to use the STRIPES index to quantify the model's ability to represent MJO teleconnection instead of using some other simpler techniques (e.g., averaging absolute values of z500 anomaly composites based on RMM phases)?

c. Discussion on potential caveats of STRIPES analysis should also be included. For example, as discussed by the authors, the propagation speed of the MJO can change with the QBO. In such a case, using the same phase speed to calculate the STRIPES index could be problematic. Is the sensitivity to choosing different phase speeds tested?

d. Line 108: Please clarify what "the resultant vector" means.

2. Section 3.1: I was a bit confused about how to interpret the results in this section. The authors explain that Figures 1 and 2 represent the sensitivity of z500 anomaly to the MJO and QBO states. However, when the authors apply the normalization, the maps appeared noisier and no regions stood out to be "sensitive" to the MJO and QBO states (in Fig. 3). Does this mean that the regions of high values in Figs. 1-2 are just regions of greater variance in z500 and do not necessarily represent the high sensitivity to the MJO and QBO? I suggest the authors recreate Figs. 1 and 2 using normalized z500 anomalies (e.g., by the standard deviation of z500), which I think would be a more proper way to show the sensitivity of z500 to the MJO and QBO states.

a. And please clarify what "distinct stripes" on line 176 and "stripey-ness" on line 181

mean.

3. Section 3.2: There were many interesting results presented in this section, but some interpretations of the results must be done more carefully. One of the conclusions that the authors make is that the prediction skills increase during MJO active states when combined with WQBO more than with EQBO states (section 3.2.4). This could be because there is a greater difference in the MJO amplitude between its active and inactive periods during WQBO then EQBO. I suggest the authors check the average amplitude of the RMM index during the different combination states of the QBO and MJO. Another point to check is if the similar samples of different RMM phases are included in each combination of QBO and MJO states. If there are any skewness in the samples of RMM phases, that should be considered for the interpretation of the results.

4. Section 3.2.5: The authors could consider eliminating this section. I am not sure how much value is added by including this section. The general finding that is summarized in this section (i.e., no relationship between z500 sensitivity and prediction skill) could be summarized in a few sentences in the summary or conclusion section.

5. Lines 336-338: I think it would be nice to add more information/discussion on the dynamics behind the importance of WQBO state to the NAO and AR associated with the MJO that were found by these cited work (Feng and Lin 2019 and Baggett et al. 2017).

---

## Author Comment (AC1) · 27 Feb 2020

1. While I appreciate the power of the STRIPES analysis, I must point out that the first time I read the paper I did not understand at all what the authors were doing. Only after skimming Jenney et al 2019 and looking at supplemental figure 1 did I fully understand what was happening. I worry that a casual reader may be less patient. To be constructive, I suggest that supplemental figure 1 be included in the main text, and I would also suggest adding a figure of lat vs. lon Z500 with a few panels corresponding to different periods explicitly showing how the wave train leads to Z500 alternating anomalies. I realize this is already in Jenney et al but a new, at first not intuitive, index needs a certain amount of repetition. As as aside, I was surprised that the STRIPES was just as strong in the European sector as in North Pacific/ NorthAmerica.

[Figure]

I would have expected a stronger response closer to the Pacific. The ACC results also indicate that the additional predictability from the MJO is mainly in the Atlantic sector too rather than the North Pacific (Figures 4 and 5). To me this is counter-intuitive, as the MJO should immediately and directly affect the North Pacific, especially in the first few weeks, and then affect the Atlantic more weakly later on. Additional discussion would be helpful. (I can try to reason why my intuition is incorrect, but really the authors should help with this)

Response: We agree that the STRIPES index is new and may not be familiar to the reader. Therefore, as suggested, we have added supplemental Figure 1 to the main paper in Section 2.3: Methods. We have additionally added two panels of spatial z500 anomalies at lead 12 days following phase 6 and phase 2 of the MJO to additionally aid the reader in understanding STRIPES. We have also included additional text: "Mid-latitude circulations can be modified by quasi-stationary rossby waves initiated by MJO convective heating. In a phase-lead diagram (e.g. Figure 1a), these are apparent as slowly alternating-sign z500 anomalies with lead following a specific phase of the MJO. In addition, the MJO is a propagating phenomenon with a phase speed of about 5-8 days/phase. Therefore, if there is a teleconnection signal 10 days following phase 2, this signal is likely also present 5 days following phase 3 in the same region, in a composite sense. On a phase-lead diagram, this is apparent as a diagonal line or 'stripe' slanted at the phase speed of the MJO (Figure 1a). Therefore, if a region is sensitive to the MJO, we expect alternating z500 anomaly stripes approximately sloped at the average phase speed of the MJO, as in Figure 1a, which we refer to as the 'stripey-ness'. For further intuition of the phase-lead diagram, Figure 1a and 1b show composite z500 anomalies for the domain around 45\degree N and 5\degree W (marked by the white X) 12 days following phase 6 and phase 2, respectively. The value of the box in the phase-lead diagram is the same as the value plotted at the X in Figure 1b,c." (See attached Figure 1 for figure.) In regards to the STRIPES result of the North Pacific, the reviewer mentions that the Pacific and European sectors have similar STRIPES values. We hypothesize that the Atlantic and European sectors may have similar STRIPES values to the Pacific from enhanced blocking over the Atlantic and Europe following the MJO (Henderson et al. 2016) leading to more persistent stripes. This explanation has been added to Section 3.1: Extratropical Sensitivity: "Interestingly, the Pacific and Atlantic sectors have similar STRIPES values. One may expect higher STRIPES values over the Pacific since it is generally known to have a strong response to the MJO. We hypothesize that the Atlantic and European sectors may have similar STRIPES values from enhanced blocking over the Atlantic and Europe following the MJO (Henderson et al. 2016) leading to large STRIPES values." In terms of the ACC result showing additional prediction skill from the MJO in Atlantic/European sector rather than over the Pacific, this is likely because prediction skill on Week 1 timescales is already generally good over all locations, and it is on this weekly timescale that the Pacific is most strongly impacted by MJO teleconnections. Therefore, we may not expect the prediction skill to be significantly different over the Pacific for these early leads. Where we would expect the MJO to provide additional prediction skill is on longer than one week timescales. This additional explanation has been added to Section 3.2.1: "Note that forecast models already have relatively good prediction skill at one week leadtimes, and therefore, we would not expect the prediction skill to be significantly different over the Pacific following MJO versus non-MJO events for these early leads. Where we would expect the MJO to provide additional prediction skill is on longer than one week timescales."

2. Between lines 192 and 203 the authors form an argument that I don't find convincing. As this argument underlies the reset of the paper, this is a major issue. To this reviewer, the clearest evidence that the QBO can enhance MJO related prediction skill would be if the difference in ACC between EQBO/MJO and EQBO/noMJO or between WQBO/MJO and WQBO/noMJO is larger than the difference between noQBO/MJO and noQBO/noMJO. Based on supplemental table 1 it seems that this kind of comparison isn't possible due to possible contamination by the ENSO signal, though perhaps the authors could compute the mean Nino3.4 index for each composite included on supplemental table 1. If the mean Nino3.4 value for each composite is small, then

La Nina and El Nino events balance out and the net prediction skill added by ENSO is small. Instead the authors evaluate a pair of differences that only partially reflect on whether the QBO is enhancing MJO related prediction skill, but rather reflect alternately on whether there is prediction skill associated with the MJO, and separately whether is prediction skill associated with the QBO (in Figures 4-6). Unless the authors perform the test in the previous paragraph, there is no basis for this statement of the authors "When these two significances appear together, we can say that a particular strong QBO increases the impact of the MJO on midlatitude prediction skill". Stated another way, the difference EQBO/MJO minus noQBO/MJO does not reflect anything about the MJO per se. Rather it reflects skill associated with EQBO. Hence I don't find figure 6 useful, other than the fact that it shows that the QBO enhances skillful forecasts in the Atlantic sector (which is a nice result, and consistent with Garfinkel et al 2018 already cited and Boer and Hamilton 2008, but the authors interpretation is completely different). In order for Figure 6 to have any bearing on the MJO, the authors need to include an additional figure showing EQBO/noMJO minus noQBO/noMJO to which we can compare the difference shown in figure 6. If there is a significant difference between EQBO/MJO minus noQBO/MJO as compared to EQBO/noMJO minus noQBO/noMJO, then there is evidence that there is some mutual interaction between the MJO and the EQBO. The authors could then rinse and repeat for WQBO. In its present form, the authors analysis only convinces me that both the QBO or the MJO separately enhance predictability on S2S timescales in these models as compared to noQBO or noMJO.

Response: This reviewer mentions that the two types of significance the authors use are insufficient as evidence for whether the QBO can enhance MJO related prediction skill. We appreciate this comment as the authors now realize how the results, as originally posed, were confusing. In fact, we completely agree with the reviewer on what must be done to make this convincing, but realize now that some of the important steps were too quickly glossed over since they ultimately had little impact on the result. With this in mind, we have now rewritten the results section on prediction skill, and in

the process, added some additional analysis to make the argument even stronger in a statistical sense. Given the statistical tests added, the resulting figures have changed - however, the overall story remains the same as requirements 1-3 were already considered/included in the earlier version of the paper. We want to thank this reviewer for their insightful comments to help us improve the heart of this paper. Specifically, we now have added 3 "requirements" that can hopefully now be more clearly stated and followed throughout the results discussion and figures. The first requirement is the presence of an MJO impact on midlatitude prediction skill during specific phases of the QBO, where an 'MJO impact' on midlatitude prediction skill is defined as a significant difference in midlatitude ACC between active MJO and inactive MJO events. The second requirement is that the magnitude of the significant MJO impact under strong QBOs is significantly larger than the significant MJO impact under NQBO. As also highlighted by the reviewer, the second requirement is calculated through a comparison of the MJO impact during strong QBOs to the MJO impact during NQBO. These two requirements together ensure that (1) there is an MJO impact and (2) that this impact is enhanced during strong QBOs compared to neutral QBOs. The third requirement is the presence of regions/leads where E/WQBO-MJO events significantly lead to higher prediction skill than NQBO-MJO given requirement 1 and 2 are satisfied. We applied this requirement to see if regions with enhanced MJO impacts during strong QBOs also have overall greater prediction skill following active MJO events compared to NQBO-MJO events, as regions of enhanced prediction skill is the focus of this paper. The reviewer also points out the small sample size of NQBO-noMJO. We agree the sample size is small and while there is not much that can be done about it, our new results include significance tests at every step that take into account the small sample sizes. We include the following discussion in the paper: "It should be noted that inactive MJOs during NQBO events with ENSO removed only occur 12 times in ECMWF and 3 times in NCEP. If ENSO events are not removed, the sample sizes increase to 47 and 52, for ECMWF and NCEP respectively (see Table S1). While there is shading across all longitudes when ENSO is removed (Figure S4), when we calculate the MJO impact

during NQBO when ENSO is included (Figure S5), we see that much of the shading east of 0\degree is not apparent. The presence of skill east of 0\degree when ENSO is not included may be due to small sample sizes of the NQBO events. Thus, when comparing MJO impacts between strong and neutral QBOs, it is important to keep sample size in mind. That being said, the statistical analysis we have applied here for requirements 1-3 account for the small sample sizes in the analysis."

3. I found section 3.2.5 extraneous and hard to understand without first skimming Tseng et al 2018. Consider deleting. Response: Thank you for this comment. The other reviewer had similar concerns and so we have decided to remove Section 3.2.5: Northern Hemisphere Prediction Skill and Sensitivity.

Minor comments: Line 13 "7-14 days", actually there is enhanced predictability up to day 28 in figures 4-6. Why limit to 14 days? Response: Thank you for pointing this out. It was an oversight, and we have updated it to say "7-28 days".

Line 77 There is earlier work that argues that the QBO may modulate ENSO teleconnections. See Garfinkel and Hartmann 2010, Richter et al 2015, and Hansen et al 2016 Response: We have added QBO effects on ENSO teleconnections to the ENSO discussion in the methods. "Some earlier research indicates that ENSO has a limited impact on the QBO-MJO interaction (e.g. \citealt{Yoo2016, Nishimoto2017}); however, recent work on QBO-MJO teleconnections has shown a possible dependency of results on ENSO \citep{Son2017, Wang2018, Sun2019}. In addition, other research suggests that the QBO affects ENSO teleconnections \citep{Garfinkel2010, Richter2015, Hansen2016}, which may consequently impact the MJO and its teleconnections."

Technical comments: Line 2 stationary Rossby wave **and** tropical-extratropical teleconnections Response: The quasi-stationary rossby waves are tropical-extratropical teleconnections, which is why we have a comma instead of 'and'.

Line 19 excitation of **quasi**stationary Rossby waves (the MJO can't force stationary waves on monthly mean or seasonal mean timescales) Response: Fixed. Thank you.

Line 126 the reference to figure 3 seems incorrect. Figure 3 shows something else entirely. Response: This should say Supplemental Figure 3. This has been corrected. Thank you.

Figure 1, title of bottom-right panel is incorrect (It probably should be WQBO-MJO) Response: Fixed. Thank you.

Please also note the supplement to this comment:
http://www.weather-clim-dynam-discuss.net/wcd-2019-13/wcd-2019-13-AC1-supplement.pdf
* * *
[Figure]

[Figure]

[Figure]

**Figure 1.** (a) Boreal winter (DJF) composite ERA-I z500 anomalies subsampled to ECMWF initialization dates (1995-2016) for each MJO phase during EQBO vs lead at 45N and 5W. White boxes and text denote corresponding figure in bottom panel. The bottom panel includes composite ERA-I z500 anomalies subsampled to ECMWF initialization dates (DJF, 1995-2016) over Europe for (b) Phase 6 and (c) Phase 2 at lead day 12. The white X denotes 45°N and 5°W.

**Fig. 1.**

---

## Author Comment (AC2) · 27 Feb 2020

1. Section 2.3: I suggest the authors revise the method section to make it more accessible to a broader audience. The authors also jump into explaining the details of each analysis technique (i.e., STRIPES and ACC). Before jumping into the details, it would be helpful to the readers if the authors could first outline what they attempt to quantify and how it relates to the objective of this study. More specifically, I suggest the following points.

a. For readers who are unfamiliar with Jenney et al. 2019, it would be difficult to understand the STRIPES index. I suggest to move the Supplemental Figure S1 to the main manuscript and include further visual illustrations on how the STRIPES index is

calculated.

Response: We agree that the STRIPES index is new and may not be familiar to the reader. Therefore, as suggested, we have added supplemental Figure 1 to the main paper in Section 2.3: Methods. We have additionally added two panels of spatial z500 anomalies at lead 12 days following phase 6 and phase 2 of the MJO to additionally aid the reader in understanding STRIPES. We have also included additional text: "Mid-latitude circulations can be modified by quasi-stationary rossby waves initiated by MJO convective heating. In a phase-lead diagram (e.g. Figure 1a), these are apparent as slowly alternating-sign z500 anomalies with lead following a specific phase of the MJO. In addition, the MJO is a propagating phenomenon with a phase speed of about 5-8 days/phase. Therefore, if there is a teleconnection signal 10 days following phase 2, this signal is likely also present 5 days following phase 3 in the same region, in a composite sense. On a phase-lead diagram, this is apparent as a diagonal line or 'stripe' slanted at the phase speed of the MJO (Figure 1a). Therefore, if a region is sensitive to the MJO, we expect alternating z500 anomaly stripes approximately sloped at the average phase speed of the MJO, as in Figure 1a, which we refer to as the 'stripey-ness'. For further intuition of the phase-lead diagram, Figure 1a and 1b show composite z500 anomalies for the domain around 45\degree N and 5\degree W (marked by the white X) 12 days following phase 6 and phase 2, respectively. The value of the box in the phase-lead diagram is the same as the value plotted at the X in Figure 1b,c."

b. I suggest the authors add more discussion on the novelty and benefits of STRIPES analysis. Why do the authors choose to use the STRIPES index to quantify the model's ability to represent MJO teleconnection instead of using some other simpler techniques (e.g., averaging absolute values of z500 anomaly composites based on RMM phases)?

Response: The STRIPES index was used over more common techniques because it allows us to regionally quantify the strength, consistency and propagation of the MJO impact on the extratropics using only one metric. This has been added to the text: "Therefore, the STRIPES index allows us to regionally quantify the strength, consistency and propagation of the MJO impact on the extratropics and thus, allows us to quantify the ability of hindcast models to capture tropical-extratropical teleconnections on one to four week timescales in a single metric."

c. Discussion on potential caveats of STRIPES analysis should also be included. For example, as discussed by the authors, the propagation speed of the MJO can change with the QBO. In such a case, using the same phase speed to calculate the STRIPES index could be problematic. Is the sensitivity to choosing different phase speeds tested?

Response: As the reviewer suggests, the changes in phase speed of MJO under different phases of the QBO may impact the STRIPES values. This is also discussed in Jenney et al. (2019) if the reviewer is interested in further discussion. Specific to this work, we conducted a sensitivity analysis and found that our STRIPES analysis and conclusions are not sensitive to the exact value of the phase speed over the range of observed phase speeds of 5-8 days/phase. This analysis of the sensitivity of the STRIPES index to the phase speed of the MJO is now included in the text: "It should be noted that the westerly phase of the QBO reduces the propagation speed of the MJO (Nishimoto and Yoden 2017), however, we find that our results are robust to changes in phase speed of +/- 2 days/phase."

d. Line 108: Please clarify what "the resultant vector" means.

Response: We have removed the term 'resultant vector', and replaced the sentence with a more detailed description. "Averages along the slopes corresponding to the MJO phase speed are calculated, and if there are alternating stripes (i.e. sensitivity to the MJO), the resulting averages concatenated into a vector will look like a sine wave, for which the amplitude can be calculated.The amplitude of this oscillatory vector is the STRIPES index (Jenney et al. 2019)."

2. Section 3.1: I was a bit confused about how to interpret the results in this section. The authors explain that Figures 1 and 2 represent the sensitivity of z500 anomaly to

the MJO and QBO states. However, when the authors apply the normalization, the maps appeared noisier and no regions stood out to be "sensitive" to the MJO and QBO states (in Fig. 3). Does this mean that the regions of high values in Figs. 1-2 are just regions of greater variance in z500 and do not necessarily represent the high sensitivity to the MJO and QBO? I suggest the authors recreate Figs. 1 and 2 using normalized z500 anomalies (e.g., by the standard deviation of z500), which I think would be a more proper way to show the sensitivity of z500 to the MJO and QBO states.

Response: We do not standardize the z500 anomalies in Figure 1, 2 and 3 because the variance of z500 has greater variability in the midlatitudes compared to the tropics and therefore, may mute the extratropical signal. In section 2.3, we state: "Since our application focuses on extratropical sensitivity in z500, we do not standardize our data for STRIPES as in Jenney et al. (2019). Standardization may mute the extratropical signal due to the greater variability of z500 in the midlatitudes, which is of main interest here. In addition, we wish to retain any differences in z500 anomaly amplitudes between the QBO phases." Furthermore, differences in composite anomaly amplitude between EQBO and WQBO are also of interest for this work. If we normalize the EQBO and WQBO by their respective maximum anomaly amplitudes in the original Figure 1 and 2 (results shown in the original Figure 3), we ignore this potential difference between the two QBO phases (i.e. one phase could lead to stronger anomalies, in a mean or event-by-event sense, than the other). The fact that the normalized plots look different compared to the non-normalized plots suggests that this anomaly amplitude difference may be appreciable between the two QBO phases, and thus, we choose not to normalize here. In regards to the standardization technique used for Figure 3 (now Figure S2), the reviewer mentions the noisiness of the figure and lack of specific regions 'sensitive' to the MJO. Since we divided by the absolute max of the z500 anomalies to normalize, the noisiness suggests the importance of the combined influence of the magnitude of the z500 anomaly as well as the stripy-ness to determine regions of sensitivity. Furthermore, the maximum is itself a noisy value. Due to the extensive confusion from this figure, and the fact that it is not a main part of this paper's focus, we have moved it to

supplemental material.

a. And please clarify what "distinct stripes" on line 176 and "stripey-ness" on line 181 mean. Response: We have added a more detailed description of distinct stripes and stripey-ness: "In addition, the MJO is a propagating phenomenon with a phase speed of approximately 5-8 days/phase. Therefore, if there is a teleconnection signal 10 days following phase 2, this signal is likely also present 5 days following phase 3 in the same region, in a composite sense. On a phase-lead diagram, this is apparent as a diagonal line or 'stripe' slanted at the phase speed of the MJO (Figure 1a). Therefore, if a region is sensitive to the MJO, we expect alternating z500 anomaly stripes approximately sloped at the average phase speed of the MJO, as in Figure 1a, which we refer to as the 'stripey-ness'."

3. Section 3.2: There were many interesting results presented in this section, but some interpretations of the results must be done more carefully. One of the conclusions that the authors make is that the prediction skills increase during MJO active states when combined with WQBO more than with EQBO states (section 3.2.4). This could be because there is a greater difference in the MJO amplitude between its active and inactive periods during WQBO then EQBO. I suggest the authors check the average amplitude of the RMM index during the different combination states of the QBO and MJO. Another point to check is if the similar samples of different RMM phases are included in each combination of QBO and MJO states. If there are any skewness in the samples of RMM phases, that should be considered for the interpretation of the Results.

Response: The reviewer suggests that the more prevalent enhanced prediction skill following active MJOs during WQBO over EQBO may be due to the differences in MJO amplitude, and suggest that the authors look at the RMM index. This is a great suggestion, and a few recent studies have found that the amplitude of the MJO is enhanced during EQBO compared to WQBO (e.g. Son et al. 2017, Nishimoto and Yoden 2017, Densmoore et al. 2019) while another says that EQBO has a greater

number of strong MJOs than WQBO (Zhang and Zhang 2018). Neither findings explain why WQBO-MJO appears to impact the midlatitude prediction skill more than EQBO-MJO. The reviewer also suggests that we check the skewness of samples of MJO phases within the analysis. This has also been calculated in Zhang and Zhang (2018), where they found that the MJO tends to propagate further into the Pacific Ocean during EQBO. However, this also does not explain why WQBO-MJO appears to impact the midlatitude prediction skill more than EQBO-MJO. With all of this said, this paper is specifically about the resulting changes in prediction skill under different QBO-MJO states, rather than a dynamical explanation behind the changes in prediction skill. This is an important next step for this work.

4. Section 3.2.5: The authors could consider eliminating this section. I am not sure how much value is added by including this section. The general finding that is summarized in this section (i.e., no relationship between z500 sensitivity and prediction skill) could be summarized in a few sentences in the summary or conclusion section.

Response: Thank you for this comment. The other reviewer had similar concerns and so we have decided to remove Section 3.2.5: Northern Hemisphere Prediction Skill and Sensitivity.

5. Lines 336-338: I think it would be nice to add more information/discussion on the dynamics behind the importance of WQBO state to the NAO and AR associated with the MJO

Response: The dynamics behind the importance of WQBO-MJO connection on the NAO and ARs is on going research. We agree this would be an interesting discussion, and an important next step. In the introduction, we hypothesize that the QBO may impact ARs through "its modulation of MJO-induced Rossby waves, and consequently, changes in the steering and frequency of atmospheric rivers." However, the paper specifically focuses on the resulting changes in prediction skill rather than the dynamical explanation behind these changes in prediction skill, and therefore, is

beyond the scope of the paper.

Please also note the supplement to this comment:
http://www.weather-clim-dynam-discuss.net/wcd-2019-13/wcd-2019-13-AC2-supplement.pdf

———————————————

[Figure]

[Figure]

[Figure]

**Figure 1.** (a) Boreal winter (DJF) composite ERA-I z500 anomalies subsampled to ECMWF initialization dates (1995-2016) for each MJO phase during EQBO vs lead at 45N and 5W. White boxes and text denote corresponding figure in bottom panel. The bottom panel includes composite ERA-I z500 anomalies subsampled to ECMWF initialization dates (DJF, 1995-2016) over Europe for (b) Phase 6 and (c) Phase 2 at lead day 12. The white X denotes 45°N and 5°W.

**Fig. 1.**

---

## Author Response (AR1)

**Major Comments:**

- Comment: While I appreciate the power of the STRIPES analysis, I must point out that the first time I read the paper I did not understand at all what the authors were doing. Only after skimming Jenney et al 2019 and looking at supplemental figure 1 did I fully understand what was happening. I worry that a casual reader may be less patient. To be constructive, I suggest that supplemental figure 1 be included in the main text, and I would also suggest adding a figure of lat vs. lon Z500 with a few panels corresponding to different periods explicitly showing how the wave train leads to Z500 alternating anomalies. I realize this is already in Jenney et al but a new, at first not intuitive, index needs a certain amount of repetition. As as aside, I was surprised that the STRIPES was just as strong in the European sector as in North Pacific/ NorthAmerica. I would have expected a stronger response closer to the Pacific. The ACC results also indicate that the additional predictability from the MJO is mainly in the Atlantic sector too rather than the North Pacific (Figures 4 and 5). To me this is counter-intuitive, as the MJO should immediately and directly affect the North Pacific, especially in the first few weeks, and then affect the Atlantic more weakly later on. Additional discussion would be helpful. (I can try to reason why my intuition is incorrect, but really the authors should help with this)
  - (a) Response:

20

25

30

35

40

45

We agree that the STRIPES index is new and may not be familiar to the reader. Therefore, as suggested, we have added supplemental Figure 1 to the main paper in Section 2.3: Methods. We have additionally added two panels of spatial z500 anomalies at lead 12 days following phase 6 and phase 2 of the MJO to additionally aid the reader in understanding STRIPES.

We have also included additional text: "... Specifically, a composite of average z500 anomalies for each MJO phase and lead (phase-lead diagram) is created for each grid point in the Northern Hemisphere (example shown in Figure 1a). For further intuition of the phase-lead diagram, Figure 1a and 1b show composite z500 anomalies for the domain around 45°N and 5°W (marked by the white X) 12 days following phase 6 and phase 2, respectively. The value of the box in the phase-lead diagram is the same as the value plotted at the X in Figure 1b,c. In a phase-lead diagram, MJO induced quasi-stationary rossby waves are apparent as slowly alternating-sign z500 anomalies with lead following a specific phase of the MJO (e.g. Figure 1a). In addition, the MJO is a propagating phenomenon with a phase speed of approximately 5-8 days/phase. Therefore, if there is a teleconnection signal 10 days following phase 2, this signal is likely also present 5 days following phase 3 in the same region, in a composite sense. On a phase-lead diagram, this is seen as a diagonal line or 'stripe' slanted at the phase speed of the MJO (Figure 1a). Therefore, if a region is sensitive to the MJO, we expect alternating z500 anomaly stripes approximately sloped at the average phase speed of the MJO, as in Figure 1a, which we refer to as the 'stripey-ness'.

To calculate STRIPES, averages along the slopes in the phase-lead diagram corresponding to the MJO phase speed are calculated, and if there are alternating stripes (i.e. sensitivity to the MJO), the resulting averages concatenated together will oscillate between positive and negative z500 anomalies as a sine wave, for which the amplitude can be calculated. The amplitude of this oscillatory vector is the STRIPES index (Jenney et al. 2019)."

In regards to the STRIPES result of the North Pacific, the reviewer mentions that the Pacific and European sectors have similar STRIPES values. We hypothesize that the Atlantic and European sectors may have similar STRIPES values to the Pacific from enhanced blocking over the Atlantic and Europe following the MJO (Henderson et al. 2016) leading to more persistent stripes.

This explanation has been added to Section 3.1: Extratropical Sensitivity: "Interestingly, the Pacific and Atlantic sectors have similar STRIPES values. One may expect higher STRIPES values over the Pacific compared to the Atlantic since the Pacific is generally known to have a strong response to the MJO. We hypothesize that the Atlantic and European sectors also have similar STRIPES values to that of the Pacific due to enhanced blocking over the Atlantic and Europe at later leads following the MJO (Henderson et al. 2016). Since the STRIPES index accounts

1

for all leads as well as the strength and consistency of the z500 anomalies, we therefore may expect STRIPES values over the Atlantic and European sectors to be large as well."

In terms of the ACC result showing additional prediction skill from the MJO in Atlantic/European sector rather than over the Pacific, this is likely because prediction skill on Week 1 timescales is already generally good over all locations, and it is on this weekly timescale that the Pacific is most strongly impacted by MJO teleconnections. Therefore, we may not expect the prediction skill to be significantly different over the Pacific for these early leads. Where we would expect the MJO to provide additional prediction skill is on longer than one week timescales.

This additional explanation has been added to Section 3.2.1: "Note that prediction skill at one week lead times is not likely to be significantly different following active MJOs compared to inactive MJOs since forecast models already have relatively good prediction skill for these early leads. Where we would expect the MJO to provide additional prediction skill is on timescales longer than one week."

2. Comment: Between lines 192 and 203 the authors form an argument that I don't find convincing. As this argument underlies the reset of the paper, this is a major issue. To this reviewer, the clearest evidence that the OBO can enhance MJO related prediction skill would be if the difference in ACC between EOBO/MJO and EOBO/noMJO or between WQBO/MJO and WQBO/noMJO is larger than the difference between noQBO/MJO and noQBO/noMJO. Based on 60 supplemental table 1 it seems that this kind of comparison isn't possible due to possible contamination by the ENSO signal, though perhaps the authors could compute the mean Nino3.4 index for each composite included on supplemental table 1. If the mean Nino3.4 value for each composite is small, then La Nina and El Nino events balance out and the net prediction skill added by ENSO is small. Instead the authors evaluate a pair of differences that only partially reflect on whether the OBO is enhancing MJO related prediction skill, but rather reflect alternately on whether there is prediction 65 skill associated with the MJO, and separately whether is prediction skill associated with the OBO (in Figures 4-6). Unless the authors perform the test in the previous paragraph, there is no basis for this statement of the authors "When these two significances appear together, we can say that a particular strong OBO increases the impact of the MJO on midlatitude prediction skill". Stated another way, the difference EQBO/MJO minus noQBO/MJO does not reflect anything about the 70 MJO per se. Rather it reflects skill associated with EQBO. Hence I don't find figure 6 useful, other than the fact that it shows that the OBO enhances skillful forecasts in the Atlantic sector (which is a nice result, and consistent with Garfinkel et al 2018 already cited and Boer and Hamilton 2008, but the authors interpretation is completely different). In order for Figure 6 to have any bearing on the MJO, the authors need to include an additional figure showing EOBO/noMJO minus noQBO/noMJO to which we can compare the difference shown in figure 6. If there is a significant difference between 75 EQBO/MJO minus noQBO/MJO as compared to EQBO/noMJO minus noQBO/noMJO, then there is evidence that there is some mutual interaction between the MJO and the EQBO. The authors could then rinse and repeat for WQBO. In its present form, the authors analysis only convinces me that both the QBO or the MJO separately enhance predictability on S2S timescales in these models as compared to noQBO or noMJO.

**(a) Response:**

50

55

90

This reviewer mentions that the two types of significance the authors use are insufficient as evidence for whether 80 the QBO can enhance MJO related prediction skill. We appreciate this comment as the authors now realize how the results, as originally posed, were confusing. In fact, we completely agree with the reviewer on what must be done to make this convincing, but realize now that some of the important steps were too quickly glossed over since they ultimately had little impact on the result. With this in mind, we have now rewritten the results section on prediction skill, and in the process, added some additional analysis to make the argument even stronger in a statistical sense. 85 Given the statistical tests added, the resulting figures have changed - however, the overall story remains the same as requirements 1-3 were already considered/included in the earlier version of the paper. We want to thank this reviewer for their insightful comments to help us improve the heart of this paper. Specifically, we now have added 3 "requirements" that can hopefully now be more clearly stated and followed throughout the results discussion and figures. The first requirement is the presence of an MJO impact on midlatitude prediction skill during specific phases of the QBO, where an 'MJO impact' on midlatitude prediction skill is defined as a significant difference in midlatitude ACC between active MJO and inactive MJO events. The second requirement is that the magnitude

of the significant MJO impact under strong QBOs is significantly larger than the significant MJO impact under NOBO. As also highlighted by the reviewer, the second requirement is calculated through a comparison of the MJO impact during strong OBOs to the MJO impact during NOBO. These two requirements together ensure that 95 (1) there is an MJO impact and (2) that this impact is enhanced during strong OBOs compared to neutral OBOs. The third requirement is the presence of regions/leads where E/WOBO-MJO events significantly lead to higher prediction skill than NOBO-MJO given requirement 1 and 2 are satisfied. We applied this requirement to see if regions with enhanced MJO impacts during strong OBOs also have overall greater prediction skill following active MJO events compared to NQBO-MJO events, as regions of enhanced prediction skill is the focus of this paper. 100 The reviewer also points out the small sample size of NQBO-noMJO. We agree the sample size is small and while there is not much that can be done about it, our new results include significance tests at every step that take into account the small sample sizes. We include the following discussion in the paper: "It should be noted that inactive MJOs during NOBO events with ENSO removed only occur 12 times in ECMWF and 3 times in NCEP. When this 105 is the case, there is shading across all longitudes (Figure S5). If ENSO events are not removed, the sample sizes increase to 47 and 52, for ECMWF and NCEP respectively (see Table S1). When we calculate the MJO impact during NOBO when ENSO is included (Figure S6), we see that much of the shading east of  $0^{\circ}$  is not apparent. The presence of skill east of 0° when ENSO is not included may be due to small sample sizes of the NOBO events. Thus, when comparing MJO impacts between strong and neutral OBOs, it is important to keep sample size in mind. That being said, the statistical analysis we have applied here for requirements 1-3 account for the small sample sizes in 110 the analysis."

- 3. Comment: I found section 3.2.5 extraneous and hard to understand without first skimming Tseng et al 2018. Consider deleting.
  - (a) Response: Thank you for this comment. The other reviewer had similar concerns and so we have decided to remove Section 3.2.5: Northern Hemisphere Prediction Skill and Sensitivity.

**Minor comments:**

115

125

130

- 4. Comment: Line 13 "7-14 days", actually there is enhanced predictability up to day 28 in figures 4-6. Why limit to 14 days?
  - (a) Response: Thank you for pointing this out. It was an oversight, and we have updated it to say "Week 1-4".
- Comment: Line 77 There is earlier work that argues that the QBO may modulate ENSO teleconnections. See Garfinkel and Hartmann 2010, Richter et al 2015, and Hansen et al 2016
  - (a) Response: We have added QBO effects on ENSO teleconnections to the ENSO discussion in the methods. "Some earlier research indicates that ENSO has a limited impact on the QBO-MJO interaction (e.g. Yoo and Son 2016; Nishimoto and Yoden 2017); however, recent work on QBO-MJO teleconnections has shown a possible dependency of results on ENSO (Son et al., 2017; Wang et al., 2018; Sun et al., 2019). In addition, other research suggests that the QBO affects ENSO teleconnections (Garfinkel and Hartmann, 2010; Richter et al., 2015; Hansen et al., 2016), which may consequently impact the MJO and its teleconnections."

**Technical comments:**

- 6. Comment: Line 2 stationary Rossby wave \*\*and\*\* tropical-extratropical teleconnections
- (a) Response: Thank you for this comment. We have changed this sentence to say: *"The Madden-Julian Oscilla-tion (MJO) is known to force extratropical weather days-to-weeks following an MJO event through excitation of stationary Rossby waves, also referred to as tropical-extratropical teleconnections."*
  - 7. Comment: Line 19 excitation of \*\*quasi\*\*stationary Rossby waves (the MJO can't force stationary waves on monthly mean or seasonal mean timescales)
    - 3

135 (

145

- (a) Response: Fixed. Thank you.
- 8. Comment: Line 126 the reference to figure 3 seems incorrect. Figure 3 shows something else entirely.
  - (a) Response: This should say Supplemental Figure 3. This has been corrected. Thank you.
- 9. Comment: Figure 1, title of bottom-right panel is incorrect (It probably should be WQBO-MJO)
  - (a) Response: Fixed. Thank you.

**140 **Response to Anonymous Referee # 2**

1. Section 2.3: I suggest the authors revise the method section to make it more accessible to a broader audience. The authors also jump into explaining the details of each analysis technique (i.e., STRIPES and ACC). Before jumping into the details, it would be helpful to the readers if the authors could first outline what they attempt to quantify and how it relates to the objective of this study. More specifically, I suggest the following points.

1.1) Comment: For readers who are unfamiliar with Jenney et al. 2019, it would be difficult to understand the STRIPES index. I suggest to move the Supplemental Figure S1 to the main manuscript and include further visual illustrations on how the STRIPES index is calculated.

(a) Response:

We agree that the STRIPES index is new and may not be familiar to the reader. Therefore, as suggested, we 150 have added supplemental Figure 1 to the main paper in Section 2.3: Methods. We have additionally added two panels of spatial z500 anomalies at lead 12 days following phase 6 and phase 2 of the MJO to additionally aid the reader in understanding STRIPES. We have also included additional text: "... Specifically, a composite of average z500 anomalies for each MJO phase and lead (phase-lead diagram) is created for each grid point in the Northern Hemisphere (example shown in Figure 1a). For further intuition of the phase-lead diagram, Figure 1a and 1b show 155 composite z500 anomalies for the domain around  $45^{\circ}$ N and  $5^{\circ}$ W (marked by the white X) 12 days following phase 6 and phase 2, respectively. The value of the box in the phase-lead diagram is the same as the value plotted at the X in Figure 1b,c. In a phase-lead diagram, MJO induced quasi-stationary rossby waves are apparent as slowly alternating-sign z500 anomalies with lead following a specific phase of the MJO (e.g. Figure 1a). In addition, the MJO is a propagating phenomenon with a phase speed of approximately 5-8 days/phase. Therefore, if there is a 160 teleconnection signal 10 days following phase 2, this signal is likely also present 5 days following phase 3 in the same region, in a composite sense. On a phase-lead diagram, this is seen as a diagonal line or 'stripe' slanted at the phase speed of the MJO (Figure 1a). Therefore, if a region is sensitive to the MJO, we expect alternating z500 anomaly stripes approximately sloped at the average phase speed of the MJO, as in Figure 1a, which we refer to as the 'stripey-ness'. 165

To calculate STRIPES, averages along the slopes in the phase-lead diagram corresponding to the MJO phase speed are calculated, and if there are alternating stripes (i.e. sensitivity to the MJO), the resulting averages concatenated together will oscillate between positive and negative z500 anomalies as a sine wave, for which the amplitude can be calculated. The amplitude of this oscillatory vector is the STRIPES index (Jenney et al. 2019)."

170 1.2) Comment: I suggest the authors add more discussion on the novelty and benefits of STRIPES analysis. Why do the authors choose to use the STRIPES index to quantify the model's ability to represent MJO teleconnection instead of using some other simpler techniques (e.g., averaging absolute values of z500 anomaly composites based on RMM phases)?

(a) Response:

175

185

190

195

210

215

The STRIPES index was used over more common techniques because it allows us to regionally quantify the strength, consistency and propagation of the MJO impact on the extratropics using only one metric. This has been added to the text: "Therefore, the STRIPES index allows us to regionally quantify the strength, consistency and propagation of the MJO impact on the extratropics and thus, allows us to quantify the ability of hindcast models to capture tropical-extratropical teleconnections on one to four week timescales in a single metric."

180 1.3) Comment: Discussion on potential caveats of STRIPES analysis should also be included. For example, as discussed by the authors, the propagation speed of the MJO can change with the QBO. In such a case, using the same phase speed to calculate the STRIPES index could be problematic. Is the sensitivity to choosing different phase speeds tested?

(a) Response:

As the reviewer suggests, the changes in phase speed of MJO under different phases of the QBO may impact the STRIPES values. This is also discussed in Jenney et al. (2019) if the reviewer is interested in further discussion. Specific to this work, we conducted a sensitivity analysis and found that our STRIPES analysis and conclusions are not sensitive to the exact value of the phase speed over the range of observed phase speeds of 5-8 days/phase. This analysis of the sensitivity of the STRIPES index to the phase speed of the MJO is now included in the text: "It should be noted that the westerly phase of the QBO has been documented to reduce the propagation speed of the MJO (Nishimoto and Yoden 2017), however, we find that our STRIPES results are robust to changes in phase speed of +/- 2 days/phase."

- 1.4) Comment: Line 108: Please clarify what "the resultant vector" means.
  - (a) Response:

We have removed the term 'resultant vector', and replaced the sentence with a more detailed description. "To calculate STRIPES, averages along the slopes in the phase-lead diagram corresponding to the MJO phase speed are calculated, and if there are alternating stripes (i.e. sensitivity to the MJO), the resulting averages concatenated together will oscillate between positive and negative z500 anomalies as a sine wave, for which the amplitude can be calculated. The amplitude of this oscillatory vector is the STRIPES index (Jenney et al. 2019)."

Section 3.1: I was a bit confused about how to interpret the results in this section. The authors explain that Figures 1 and 2 represent the sensitivity of z500 anomaly to the MJO and QBO states. However, when the authors apply the normalization, the maps appeared noisier and no regions stood out to be "sensitive" to the MJO and QBO states (in Fig. 3). Does this mean that the regions of high values in Figs. 1-2 are just regions of greater variance in z500 and do not necessarily represent the high sensitivity to the MJO and QBO? I suggest the authors recreate Figs. 1 and 2 using normalized z500 anomalies (e.g., by the standard deviation of z500), which I think would be a more proper way to show the sensitivity of z500 to the MJO and QBO states.

(a) Response:

We do not standardize the z500 anomalies in Figure 1, 2 and 3 because the variance of z500 has greater variability in the midlatitudes compared to the tropics and therefore, may mute the extratropical signal. Furthermore, differences in composite anomaly amplitude between EQBO and WQBO are also of interest for this work. If we normalize the EQBO and WQBO by their respective maximum anomaly amplitudes in the original Figure 1 and 2 (results shown in the original Figure 3), we ignore this potential difference between the two QBO phases (i.e. one phase could lead to stronger anomalies, in a mean or event-by-event sense, than the other). In section 2.3, we state: "Also note that since our application focuses on extratropical sensitivity in z500, we use z500 anomalies in terms of meters instead of standard deviation for STRIPES, different from Jenney et al. (2019). Standardization may mute the extratropical signal due to the greater variability of z500 in the midlatitudes, which is of main interest here. In addition, we wish to retain any differences in z500 anomaly amplitudes between the QBO phases." The fact that the normalized plots look different compared to the non-normalized plots suggests that this anomaly amplitude difference may be appreciable between the two QBO phases, and thus, we choose not to normalize here. In regards to the standardization technique used for Figure 3 (now Figure S2), the reviewer mentions the noisiness of the figure and lack of specific regions 'sensitive' to the MJO. Since we divided by the absolute max of the z500 anomalies to normalize, the noisiness suggests the importance of the combined influence of the magnitude of the z500 anomaly as well as the stripy-ness to determine regions of sensitivity. Furthermore, the maximum is itself a noisy value. Due to the extensive confusion from this figure, and the fact that it is not a main part of this paper's focus, we have moved it to supplemental material.

- 225 2.1) Comment: And please clarify what "distinct stripes" on line 176 and "stripey-ness" on line 181 mean.
  - (a) Response:

220

230

We have added a more detailed description of distinct stripes and stripey-ness: "In addition, the MJO is a propagating phenomenon with a phase speed of approximately 5-8 days/phase. Therefore, if there is a teleconnection signal 10 days following phase 2, this signal is likely also present 5 days following phase 3 in the same region, in a composite sense. On a phase-lead diagram, this is seen as a diagonal line or 'stripe' slanted at the phase speed of the MJO (Figure 1a). Therefore, if a region is sensitive to the MJO, we expect alternating z500 anomaly stripes approximately sloped at the average phase speed of the MJO, as in Figure 1a, which we refer to as the 'stripey-ness'."

3. Section 3.2: There were many interesting results presented in this section, but some interpretations of the results must be done more carefully. One of the conclusions that the authors make is that the prediction skills increase during MJO active states when combined with WQBO more than with EQBO states (section 3.2.4). This could be because there is a greater difference in the MJO amplitude between its active and inactive periods during WQBO then EQBO. I suggest the authors check the average amplitude of the RMM index during the different combination states of the QBO and MJO. Another point to check is if the similar samples of different RMM phases are included in each combination of QBO and MJO states. If there are any skewness in the samples of RMM phases, that should be considered for the interpretation of the Results.

(a) Response:

The reviewer suggests that the more prevalent enhanced prediction skill following active MJOs during WQBO over EQBO may be due to the differences in MJO amplitude, and suggest that the authors look at the RMM index. This is a great suggestion, and a few recent studies have found that the amplitude of the MJO is enhanced during EQBO compared to WQBO (e.g. Son et al. 2017, Nishimoto and Yoden 2017, Densmoore et al. 2019) while another says that EQBO has a greater number of strong MJOs than WQBO (Zhang and Zhang 2018). Neither findings explain why WQBO-MJO appears to impact the midlatitude prediction skill more than EQBO-MJO. The reviewer also suggests that we check the skewness of samples of MJO phases within the analysis. This has also been calculated in Zhang and Zhang (2018), where they found that the MJO tends to propagate further into the Pacific Ocean during EQBO. However, this also does not explain why WQBO-MJO appears to impact the midlatitude prediction skill more than EQBO-MJO. With all of this said, this paper is specifically about the resulting changes in prediction skill under different QBO-MJO states, rather than a dynamical explanation behind the changes in prediction skill. This is an important next step for this work.

- 4. Section 3.2.5: The authors could consider eliminating this section. I am not sure how much value is added by including this section. The general finding that is summarized in this section (i.e., no relationship between z500 sensitivity and prediction skill) could be summarized in a few sentences in the summary or conclusion section.
  - (a) Response:

Thank you for this comment. The other reviewer had similar concerns and so we have decided to remove Section 3.2.5: Northern Hemisphere Prediction Skill and Sensitivity.

260

6

- 5. Lines 336-338: I think it would be nice to add more information/discussion on the dynamics behind the importance of WQBO state to the NAO and AR associated with the MJO
  - (a) Response:

The dynamics behind the importance of WQBO-MJO connection on the NAO and ARs is on going research. We agree this would be an interesting discussion, and an important next step. In the introduction, we hypothesize that the QBO may impact ARs through "its modulation of MJO-induced Rossby waves, and consequently, changes in the steering and frequency of atmospheric rivers." However, the paper specifically focuses on the resulting changes in prediction skill rather than the dynamical explanation behind these changes in prediction skill, and therefore, is beyond the scope of the paper.

[revised manuscript text omitted]

---

## Author Response (AR2)

**Co-Editor Decision: Publish subject to minor revisions (review by editor)** (13 Apr 2020)
by Michael Riemer

Dear Kirsten Mayer,
Both referees are very happy with your revisions and the manuscript is now suitable for publication subject to very few further minor revisions (see referee reports). An interesting point is raised by referee 1 that I believe is worth giving consideration before final publication of your manuscript.

**Reviewer 1:**
The authors have satisfactorily addressed my previous comments. I just have one final question, and though a thorough treatment is likely beyond the scope of this work as the authors acknowledge their focus isn't on mechanisms, I would appreciate some discussion.

The authors restrict their analysis to the first 28 days, but figure 5 makes it clear that skill associated with the QBO isn't decaying at the end of this period as significant anomalies are still present at day 28. In other words, it appears that the QBO leads to a **lengthening** of the period over which the MJO has a significant impact. At shorter lags the MJO teleconnection doesn't depend on the QBO phase, but at longer lags it does depend on the QBO phase. Do the results in figure 5 persist for lags beyond day 28? ECMWF has data out to day 46 if I remember correctly, and NCEP extends even longer.

Assuming my interpretation is correct, a possible mechanism for this effect is the MJO affecting the Arctic stratosphere and then subsequently affecting the surface (Schwartz and Garfinkel 2017), and in particular there seems to be some sensitivity of the type of response of the Arctic stratosphere to the MJO based on the QBO phase (Liu et al 2014). I'm not aware of anyone who has looked at the downward impact to the surface however.

Schwartz, C. and Garfinkel, C.I., 2017. Relative roles of the MJO and stratospheric variability in North Atlantic and European winter climate. Journal of Geophysical Research: Atmospheres, 122(8), pp.4184-4201.

Liu, C., Tian, B., Li, K.F., Manney, G.L., Livesey, N.J., Yung, Y.L. and Waliser, D.E., 2014. Northern Hemisphere mid-winter vortex-displacement and vortex-split stratospheric sudden warmings: Influence of the Madden-Julian Oscillation and Quasi-Biennial Oscillation. Journal of Geophysical Research: Atmospheres, 119(22), pp.12-599.

If the authors haven't already downloaded the data beyond day 28 they shouldn't start now, but it would be interesting to me to see if their results extend to days 30-45

**Response to Reviewer 1:**

Thank you for your comment.

We have the hindcast data downloaded out to lead 40 for both ECMWF and NCEP, so we extended the analysis out to 40 days (see attached Figure). The reviewer comments about a possible lengthening of the MJO impact due to strong QBOs. If this is the case, we expect requirement 1 and 2 to be satisfied on Week 5-6 timescales. As a reminder, requirement 1 (grey dots) indicates leads and longitudes where there is a significant MJO impact on prediction skill for either EQBO or WQBO (strong QBOs). Requirement 2 (black circles around grey dots), indicates where and when the MJO impact during a strong QBO is significantly greater than an MJO impact during NQBO. Requirement 3 (small black dots) indicates significantly enhanced prediction skill following MJO activity during strong QBOs compared to neutral QBOs. Passing both requirement 1 and 2 would suggest that there is an MJO impact during a strong QBO that is greater than an MJO impact during NQBO at lead times greater than 4 weeks, and thus a lengthening of the MJO impact due to strong QBOs.

We find that from week 5 into week 6 following EQBO-MJO events in ECMWF (a), requirement 1 is satisfied from the Pacific Ocean into North America. For both models, there is an MJO impact over the West Pacific at week 6 (a,b). In addition, for NCEP, there is an MJO impact following EQBO-MJO events across Eastern Europe through Asia over Week 5 (b). Requirement 2 is satisfied over the Pacific into North America in ECMWF and over Europe through Asia on NCEP during weeks 5-6. This implies that an MJO impact during EQBO is significantly greater than MJO impacts during NQBO in these regions. However, requirement 3 is not satisfied, and therefore, the prediction skill provided by the MJO during EQBO is not significantly greater than that during NQBO.

For WQBO in ECMWF, requirement 1 is satisfied over the North Atlantic during Week 5 and over North America during Week 6 (c). For NCEP, there is an MJO impact over the North Atlantic and Europe during Week 5 into 6, and an impact over western North America during Week 6 (d). There are a few points where requirements 2 and 3 are satisfied at Week 5 and 6 in ECMWF, but they are relatively incohesive. On the other hand, both requirements are satisfied over the North Atlantic during Week 5 and the Pacific during Week 6 in NCEP.

Referring back to the paper, Figure 5 suggests that WQBOs enhance the MJO impact on midlatitude prediction skill as well as enhance overall prediction skill compared to NQBOs following active MJOs from the Pacific to Europe on Week 1-4 timescales in ECMWF and on Week 3-4 timescales in NCEP. Extending the analysis out to 40 days suggests that we could possibly extend the timescales for NCEP from Week 3-4 to Week 3-6. However, it is important to note that these clusters of significance are less structured and smaller than during weeks 3-4 and there are no overlapping significant regions with ECMWF.

From this analysis, we did see that both models following strong QBO-MJOs had significant MJO impacts through Week 6 (requirement 1); however, there are only a few points

where this MJO impact was also significantly larger than the MJO impact during NQBOs (requirement 2). Therefore, it appears that the QBO may not actually lead to a lengthening of the MJO impact. With that said, it could be that the small regions that do pass our tests are related to regional stratospheric signals, but this would require additional analysis to investigate and thus is outside the scope for this paper.

We have included the following text in section 3.2.1 to address this question:
*"We also extended this analysis beyond Week 4 (not shown). We find that while there is still an MJO impact through 40 days, this MJO impact is seldom significantly larger than the MJO impact during NQBOs. This suggests that strong QBOs do not lead to an enhanced MJO impact beyond 4 weeks in these models."*

[Figure]

**Reviewer 2:**

The authors have addressed all of my previous comments. The manuscript provides interesting findings and it is easier to read now with added clarifications.

Comments

1. Line 108: "Figure 1a and 1b" should be "Figure 1b and 1c".

2. Figures 2 & 3: I suggest adding to the captions that the statistical significance is only calculated for ERA-I.

3. Figures 4 and 5: I suggest adding units (days) to the lead time.

**Response to Reviewer 2:**

1. Fixed. Thank you.
2. Thank you for this suggestion. It has been added to the captions of Figures 2 and 3.
3. We have added the units of 'days' to the lead time on Figures 4 and 5.

[revised manuscript text omitted]